# Effects of Lumacaftor-Ivacaftor on Airway Microbiota-Mycobiota and Inflammation in Patients with Cystic Fibrosis Appear To Be Linked to *Pseudomonas aeruginosa* Chronic Colonization

Raphael Enaud,[a,b,c] Florian Lussac-Sorton,[a,b,d] Elena Charpentier,[a,b,d] Lourdes Velo-Suárez,[e,f] Jennifer Guiraud,[g] Stéphanie Bui,[a,c] Michael Fayon,[a,b,c] Thierry Schaeverbeke,[b] Macha Nikolski,[h,i] the LumIvaBiota Study Group, Pierre-Régis Burgel,[j,k] Geneviève Héry-Arnaud,[e,l] Laurence Delhaes[a,b,c,d]

aBordeaux University, INSERM U1045, Centre de Recherche Cardio-thoracique de Bordeaux, Bordeaux, France
bBordeaux University Hospital, Bordeaux University, FHU ACRONIM, Bordeaux, France
cBordeaux University Hospital, CRCM Pédiatrique, CIC 1401, Bordeaux, France
dBordeaux University Hospital, Service de Parasitologie-Mycologie, Bordeaux, France
eBrest University, EFS, UMR 1078, GGB, INSERM, Brest, France
fBrest Center for Microbiota Analysis (CBAM), CHU Brest, Brest, France
gBordeaux University Hospital, Service de Bactériologie, Bordeaux, France
hBordeaux University, Bordeaux Bioinformatics Center, Bordeaux, France
iBordeaux University, CNRS, Institut de Biochimie et Génétique Cellulaires (IBGC) UMR 5095, Talence, France
jUniversité de Paris, Institut Cochin, INSERM U1016, Paris, France
kCochin Hospital, Assistance Publique-Hôpitaux de Paris (AP-HP), Respiratory Medicine and National Reference Cystic Fibrosis Reference Center, Paris, France
lBrest University Hospital, Department of Bacteriology, Virology, Hospital Hygiene, and Parasitology-Mycology, Brest, France

**ABSTRACT** Lumacaftor-ivacaftor is a cystic fibrosis transmembrane conductance regulator (CFTR) modulator combination approved for patients with cystic fibrosis (CF) who are homozygous for the F508del allele. This treatment showed significant clinical improvement; however, few studies have addressed the evolution of the airway microbiota-mycobiota and inflammation in patients receiving lumacaftor-ivacaftor treatment. Seventy-five patients with CF aged 12 years or older were enrolled at the initiation of lumacaftor-ivacaftor therapy. Among them, 41 had spontaneously produced sputa collected before and 6 months after treatment initiation. Airway microbiota and mycobiota analyses were performed via high-throughput sequencing. Airway inflammation was assessed by measuring the calprotectin levels in sputum; the microbial biomass was evaluated via quantitative PCR (qPCR). At baseline ($n = 75$), bacterial alpha-diversity was correlated with pulmonary function. After 6 months of lumacaftor-ivacaftor treatment, a significant improvement in the body mass index and a decreased number of intravenous antibiotic courses were noted. No significant changes in bacterial and fungal alpha- and beta-diversities, pathogen abundances, or calprotectin levels were observed. However, for patients not chronically colonized with *Pseudomonas aeruginosa* at treatment initiation, calprotectin levels were lower, and a significant increase in bacterial alpha-diversity was observed at 6 months. This study shows that the evolution of the airway microbiota-mycobiota in CF patients depends on the patient's characteristics at lumacaftor-ivacaftor treatment initiation, notably chronic colonization with *P. aeruginosa*.

**IMPORTANCE** The management of cystic fibrosis has been transformed recently by the advent of CFTR modulators, including lumacaftor-ivacaftor. However, the effects of such therapies on the airway ecosystem, particularly on the microbiota-mycobiota and local inflammation, which are involved in the evolution of pulmonary damage, are unclear. This multicenter study of the evolution of the microbiota under protein therapy supports the notion that CFTR modulators should be started as soon as

Address correspondence to Laurence Delhaes, laurence.delhaes@u-bordeaux.fr.

The authors declare a conflict of interest. This work was funded by research grants from VERTEX and Vaincre la Mucoviscidose association. Neither Vertex society or Vaincre La Mucoviscidose association played any role in the data collection and analysis, or in the decision to submit the article.

possible, ideally before the patient is chronically colonized with *P. aeruginosa*. (This study has been registered at ClinicalTrials.gov under identifier NCT03565692).

**KEYWORDS** cystic fibrosis, airway microbiome, airway mycobiome, airway inflammation, lumacaftor-ivacaftor, inflammation, microbiome

Cystic fibrosis (CF) is a predominant genetic disease in Caucasian populations, caused by mutations in the gene encoding the CF transmembrane conductance regulator (CFTR) protein. The corresponding defect in ion-water transport results in multiple-organ dysfunction, but airway colonization and infection, lung inflammation, and malnutrition are among the most important prognostic factors in patients with CF (1).

The lung ecosystem is now considered polymicrobial, and advances in next-generation sequencing have led to the better characterization of its bacterial (microbiota) and fungal (mycobiota) communities (2–4). In CF, reduced airway clearance, the accumulation of respiratory secretions, as well as other factors, such as the host immune response and treatments (e.g., antibiotics), contribute to the disruption of the airway microbiota-mycobiota, which includes a decrease in microbial alpha-diversity and an overrepresentation of pathogens. Such changes are associated with lung disease severity in CF patients and are a biomarker of disease evolution. Airway microbiota alpha-diversity is inversely correlated with pulmonary inflammation and positively correlated with the percent predicted forced expiratory volume in 1 s (ppFEV$_1$) and body mass index (BMI). Microbiota alpha-diversity is also predictive of disease evolution (3, 5, 6).

Cystic fibrosis management was previously limited to symptomatic management with inhaled mucolytics, airway clearance techniques, antibiotic courses to treat respiratory infections, associated pancreatic extracts, and nutritional supplements. However, in the past decade, the therapeutic management of CF patients has been changed by the development of CFTR modulators, long-term treatments aimed at restoring the functionality of the CFTR protein. In 2015, the combination of lumacaftor (LUM) and ivacaftor (IVA) (CFTR corrector and potentiator, respectively) became the first protein therapy approved for F508del homozygous patients (about 40% of patients with CF) (7, 8). Clinical trials have confirmed the efficacy and safety of lumacaftor-ivacaftor in patients with CF aged 12 years and older (9). It significantly improves the ppFEV$_1$, pulmonary exacerbations, BMI, chloride concentrations in sweat, the lung clearance index (LCI), and the parameters of magnetic resonance imaging (MRI) of the chest (8, 10, 11). However, the correlation between lung function improvement and sweat chloride levels is only modest, suggesting that the patient response to lumacaftor-ivacaftor is multifactorial (12). Moreover, the clinical response is heterogeneous during the first 6 months (13, 14).

It is therefore important to investigate the effects of modulators, in particular their effects on airway microbiology, for several reasons. First, it should be done to assess their impacts on underlying pathophysiology, with the objective of reducing the abundance of pathogens and restoring the pulmonary microbiota to one more closely resembling the microbiota in early disease (increases in commensal organisms and microbial alpha-diversity). Moreover, these findings could provide biomarkers of treatment efficacy, which could be used to adapt associated therapies (e.g., the continuation of inhaled antibiotics).

CFTR modulators can modify the airway microbiome in several ways. First, direct antimicrobial properties have been described *in vitro* for ivacaftor and lumacaftor, including against *Pseudomonas aeruginosa* and *Staphylococcus aureus* (15–17). In addition, indirect antimicrobial effects are possible, including an improvement of mucociliary clearance, modification of pH, hydration of airway secretions, and adaptation of associated therapies (such as antibiotic use). Several studies that have investigated the effects of CFTR modulator therapies on the airway microbiome have reported significant changes in composition, sometimes as early as the first weeks of treatment (11, 18–26). Some of them have shown an increase in bacterial alpha-diversity and a decrease in the abundance of *P. aeruginosa* bacteria in sputum, but these results are inconsistent with those of other reports. In addition, most studies have involved

patients with at least one G551D mutation treated with ivacaftor. Fewer studies have focused on the evolution of the airway microbiome after lumacaftor-ivacaftor therapy (11, 26). There are limited and contradictory results, and most studies have not taken into account the mycobiota and inflammation (11), both of which are biomarkers of lung function (3, 6, 18, 27, 28).

In this study, we hypothesized that improved CFTR function after lumacaftor-ivacaftor initiation might be associated with changes in the airway microbiota-mycobiota composition and decreased airway inflammation. We examined associations among the airway microbiota-mycobiota, airway inflammation, and clinical outcomes in sputum samples from a multicenter cohort of CF patients aged 12 years and older, prospectively followed up before and 6 months after the initiation of lumacaftor-ivacaftor treatment.

## RESULTS

**Patient characteristics at baseline.** Seventy-five CF patients (35 [47%] 12 to 17 years old and 40 [53%] ≥18 years old) were analyzed. Patient characteristics at baseline are summarized in Table 1. All patients received continuous lumacaftor-ivacaftor treatment and were followed up for 6 months, but only 41 (55%) had a sufficient volume of spontaneous sputa for metabarcoding analysis at the second visit (i.e., the subgroup with pairs of sputum samples at treatment initiation [M0] and at 6 months of follow-up [M6]) (see Fig. S1 in the supplemental material). At baseline, these 41 patients had significantly lower $ppFEV_1$ values and greater *P. aeruginosa* chronic colonization; no other significant differences were observed compared to the whole population (Table 1).

**Airway microbiota-mycobiota and inflammation at baseline.** Among the final median count of 30,890 bacterial reads (interquartile range [IQR], 24,815, 37,816 reads) for 1,329 bacterial amplicon sequence variants (ASVs), the microbiota composition at baseline was dominated by *Pseudomonas* (21%), followed by *Streptococcus* (13%) and *Prevotella* (12%). *Veillonella* and *Staphylococcus* each accounted for 10% of the bacterial ASVs (Fig. S2B). Among the factors influencing the microbiota composition at baseline, adults had an airway microbiota distinct from that of adolescents, as shown by the beta-diversity ($P = 0.01$ by permutational multivariate analysis of variance [PERMANOVA]) (Fig. S3A), the lower alpha-diversity (Fig. S3B), and the overrepresentation of *P. aeruginosa* (Fig. S3C). Patients with a $ppFEV_1$ of ≥80% also had a distinct microbiota in terms of alpha-diversity (Shannon and Simpson index $P$ values of 0.03 and 0.01, respectively) (Fig. 1A) and beta-diversity ($P = 0.02$) (Fig. 1B) compared to patients with a $ppFEV_1$ of <80% at baseline. Furthermore, alpha-diversity indices were correlated with $ppFEV_1$ (Fig. 1C). Using the linear discriminant analysis (LDA) effect size (LEfSe) method, we identified numerous taxonomic nodes predicting $ppFEV_1$ in our population. Patients with a $ppFEV_1$ of <80% showed significantly increased relative abundances of *Pseudomonas* and *Lautropia* (Fig. 1D). Among the 16 genera whose relative abundances were significantly different at baseline in patients with an $FEV_1$ of ≥80%, there was an overrepresentation of *Streptococcus*, *Porphyromonas*, *Actinomyces*, TM7x, and *Peptostreptococcus* (Fig. 1D).

Regarding the mycobiota, the final median count was 2,031 reads (IQR, 671, 7,722 reads) for 493 fungal ASVs. The mycobiota composition at baseline was dominated by *Candida* (35%), followed by *Malassezia* (14%) and *Saccharomyces* (8%). *Aspergillus* accounted for 5% of the fungal ASVs (Fig. S2A). Of note, 19 (25%) of the 75 patients had a positive galactomannan (GM) index, but no significant association was observed between the GM level and chronic colonization status or the relative abundance of *A. fumigatus*. The mycobiota was not significantly different in alpha- or beta-diversity according to age (adolescents versus adults) (data not shown) or lung function ($ppFEV_1$ of ≥80% versus $ppFEV_1$ of <80%) (Fig. S4).

At baseline, the mean calprotectin level in sputum was 3,941 $\mu$g/mL (IQR, 3,098, 4,581 $\mu$g/mL) (Table 1). The level significantly increased with age ($r = 0.43$; $P = 0.003$) and was negatively correlated with the $ppFEV_1$ value and bacterial alpha-diversity indices ($r = -0.48$, $-0.53$, and $-0.59$, respectively; $P < 0.001$) (Fig. 2A to D). Calprotectin levels were also significantly higher in patients chronically colonized with *P. aeruginosa*

**TABLE 1** Clinical characteristics at baseline of all patients with CF and the subgroups with and without M6 samples[a]

| Parameter | All patients No. of missing values[b] | All patients Value | Patients with M0 and M6 samples No. of missing values[b] | Patients with M0 and M6 samples Value | Patients without M6 samples No. of missing values[b] | Patients without M6 samples Value | P value[c] |
|---|---|---|---|---|---|---|---|
| No. (%) of patients | | 75 | | 41 (55) | | 34 (45) | |
| Median age (yrs) ($\pm$SD) | 0 | 21.0 ($\pm$9.0) | 0 | 22.5 ($\pm$9.0) | 0 | 19.0 ($\pm$8.7) | 0.10 |
| No. (%) of patients <18 yrs of age | | 35 (47) | | 16 (39) | | 19 (54) | 0.16 |
| No. (%) of female patients | 0 | 40 (53) | 0 | 19 (46) | 0 | 21 (62) | 0.18 |
| Median ppFEV$_1$ (%) (IQR) | 4 | 65 (53, 84) | 1 | 58 (42, 78) | 3 | 73 (62, 87) | **0.01** |
| Median BMI (kg/m$^2$) (IQR) | 2 | 18.8 (17.3, 21.0) | 2 | 19.2 (17.5, 20.3) | 0 | 18.7 (16.9, 21.6) | 0.92 |
| Median BMI Z-score (IQR)[d] | | $-0.8$ ($-1.1$, 0.1) | | $-0.8$ ($-1.2$, $-0.1$) | | $-0.8$ ($-1.1$, 0.2) | 0.59 |
| No. (%) of patients with $\geq$1 i.v. antibiotic course in the previous 12 mo | 4 | 40 (56) | 1 | 25 (64) | 3 | 15 (47) | 0.15 |
| No. (%) of patients taking maintenance pulmonary medication(s) at baseline | | | | | | | |
| Inhaled antibiotics | 1 | 45 (61) | 1 | 26 (65) | 0 | 19 (56) | 0.42 |
| Azithromycin | 0 | 45 (60) | 0 | 26 (63) | 0 | 19 (56) | 0.51 |
| Dornase alfa | 1 | 59 (80) | 1 | 31 (78) | 0 | 28 (82) | 0.6 |
| Inhaled corticosteroids | 0 | 52 (69) | 0 | 32 (78) | 0 | 20 (59) | 0.07 |
| Oral corticosteroids | 0 | 3 (4) | 0 | 2 (5) | 0 | 1 (3) | 1 |
| Inhaled hypertonic saline | 5 | 8 (11) | 5 | 3 (8) | 0 | 5 (15) | 0.47 |
| Inhaled bronchodilators | 5 | 59 (84) | 5 | 30 (83) | 0 | 29 (85) | 0.82 |
| No. (%) of patients with pulmonary colonization | | | | | | | |
| MSSA | 0 | 45 (60) | 0 | 21 (51) | 0 | 24 (71) | 0.09 |
| MRSA | 0 | 12 (16) | 0 | 8 (20) | 0 | 4 (12) | 0.36 |
| H. influenzae | 0 | 7 (9) | 0 | 3 (7) | 0 | 4 (12) | 0.7 |
| P. aeruginosa[e] | 0 | 41 (55) | 0 | 27 (66) | 0 | 14 (41) | **0.03** |
| B. cepacia | 0 | 2 (3) | 0 | 1 (2) | 0 | 1 (3) | 1 |
| A. fumigatus[e] | 16 | 23 (39) | 13 | 13 (46) | 3 | 10 (32) | 0.27 |
| Sputum supernatant dosage | | | | | | | |
| Median calprotectin level ($\mu$g/mL) (IQR) | 11 | 3,941 (3,098, 4,581) | 6 | 4,232 (3,216, 4,652) | 5 | 3,696 (2,251, 4,471) | 0.17 |
| No. (%) of patients with a GM index of >1 | 5 | 19 (25) | 1 | 13 (32) | 4 | 6 (20) | 0.24 |
| Median total fungal load (log pg/$\mu$L) (IQR) | 0 | 0.7 (0, 1.3) | 0 | 0.7 (0, 1.3) | 0 | 0.7 (0, 1.1) | 0.7 |
| Median total bacterial load (log pg/$\mu$L) (IQR) | 0 | 2.3 (1.8, 2.8) | 0 | 2.1 (1.7, 2.6) | 0 | 2.4 (2.1, 2.8) | 0.1 |
| Median P. aeruginosa load (log copies/mL) (IQR) | 1 | 3.0 (2.4, 6.3) | 1 | 3.7 (2.4, 6.1) | 0 | 2.7 (2.2, 6.7) | 0.7 |

[a]Data are means ($\pm$SD), medians (IQRs), or numbers (percentages). ppFEV$_1$, percent predicted forced expiratory volume in 1 s; BMI, body mass index; MSSA, methicillin-susceptible S. aureus; MRSA, methicillin-resistant S. aureus.
[b]Missing values correspond to the number of patients without the corresponding data.
[c]Comparisons between patients with and those without M6 samples. Significant values ($p < 0.05$) are in boldface.
[d]For adolescents.
[e]P. aeruginosa and A. fumigatus colonizations refer to chronic colonization status defined according to local practices as the presence of P. aeruginosa isolates in 3 consecutive cultures with at least 1 month between positive cultures during the previous 6 months (49) or as >50% of samples positive in the last 12 months (50) and as 2 sputum cultures positive for A. fumigatus during the last 12 months (51).

than in those not chronically colonized (4,278 $\mu$g/mL [IQR, 3,827, 4,743 $\mu$g/mL] and 3,168 $\mu$g/mL [IQR, 768, 3,902 $\mu$g/mL], respectively) ($P < 0.001$) and were correlated with P. aeruginosa loads quantification ($r = 0.44$; $P < 0.001$) (Fig. 2E and F). Furthermore, the compositions of the airway microbiota, but not the mycobiota, significantly differed in terms of alpha- and beta-diversities between patients with and those without P. aeruginosa chronic colonization (Fig. 3).

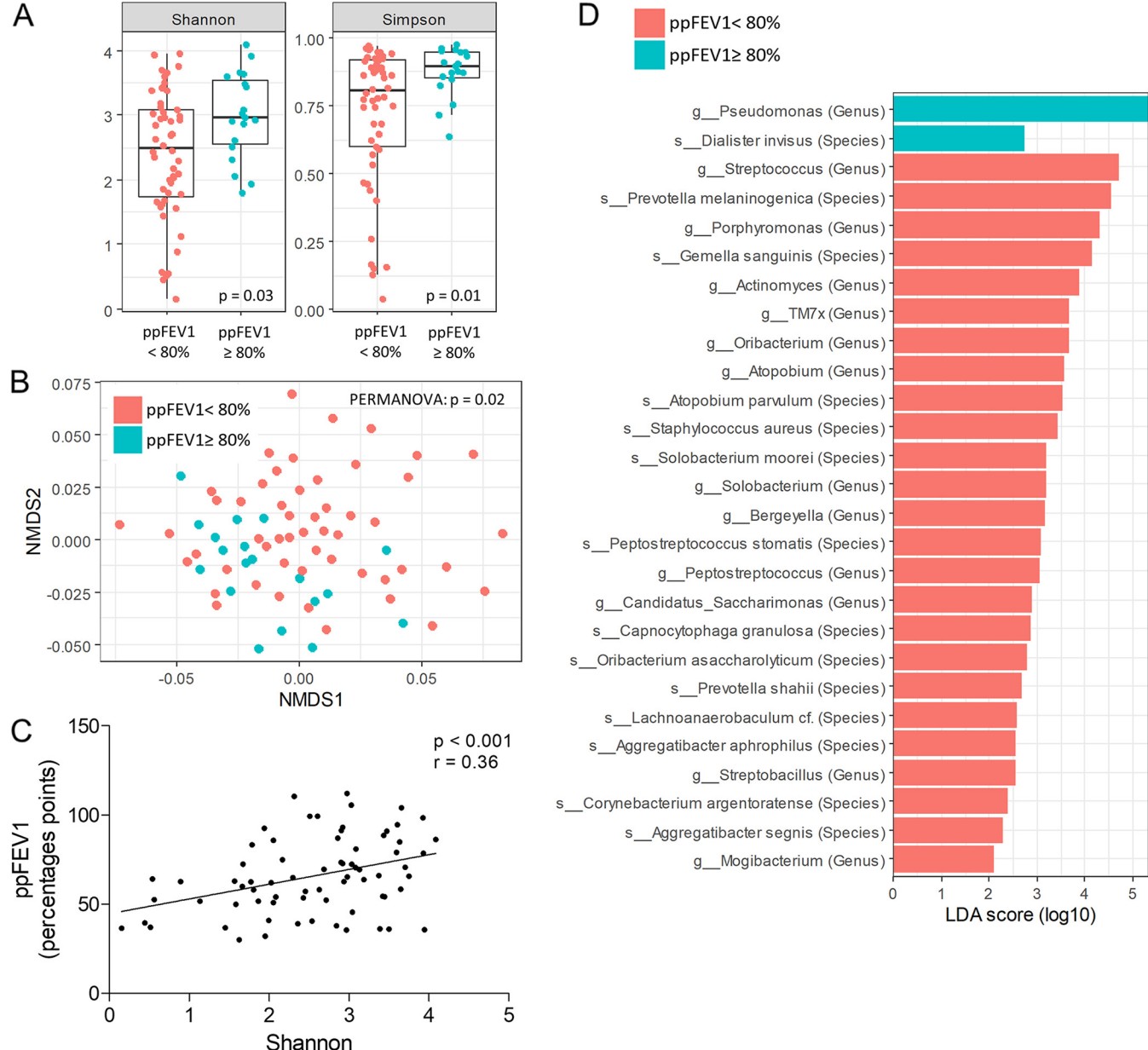

**FIG 1** Bacterial composition of sputum at baseline according to lung function. (A) Alpha-diversity indices (Shannon and Simpson) of the microbiota. (B) Beta-diversity (which assesses differences in microbiota compositions between samples) according to lung function at baseline, using a nonmetric multidimensional scaling (NMDS) ordination method with the Bray-Curtis distance metric. (C) Correlation between the ppFEV$_1$ and the Shannon index. (D) LEfSe method showing ASVs distinguishing patients with a ppFEV$_1$ of <80% and a ppFEV$_1$ of ≥80% at baseline.

**Changes in clinical outcomes, the microbiota-mycobiota, and inflammation characteristics under lumacaftor-ivacaftor treatment are linked to _P. aeruginosa_ chronic colonization status at baseline.** Changes in clinical outcomes between M0 and M6 for all patients and the subgroup of 41 patients with paired sputum samples (M0-M6 sputum samples) were comparable (Table 2). No significant changes in ppFEV$_1$ but significant increases in BMI were observed, and the BMI Z-score was stable in adolescents in both populations. Patients also had a significant decrease in the number of intravenous (i.v.) antibiotic courses (Table 2).

Next, we focused on the subgroup of 41 patients with M0-M6 sputum samples to decipher the evolution of the airway microbiota and mycobiota and sputum biomarkers. The median time between treatment initiation and the M6 visit was 183 days (IQR, 165, 210 days). The airway microbiota and mycobiota did not have significantly

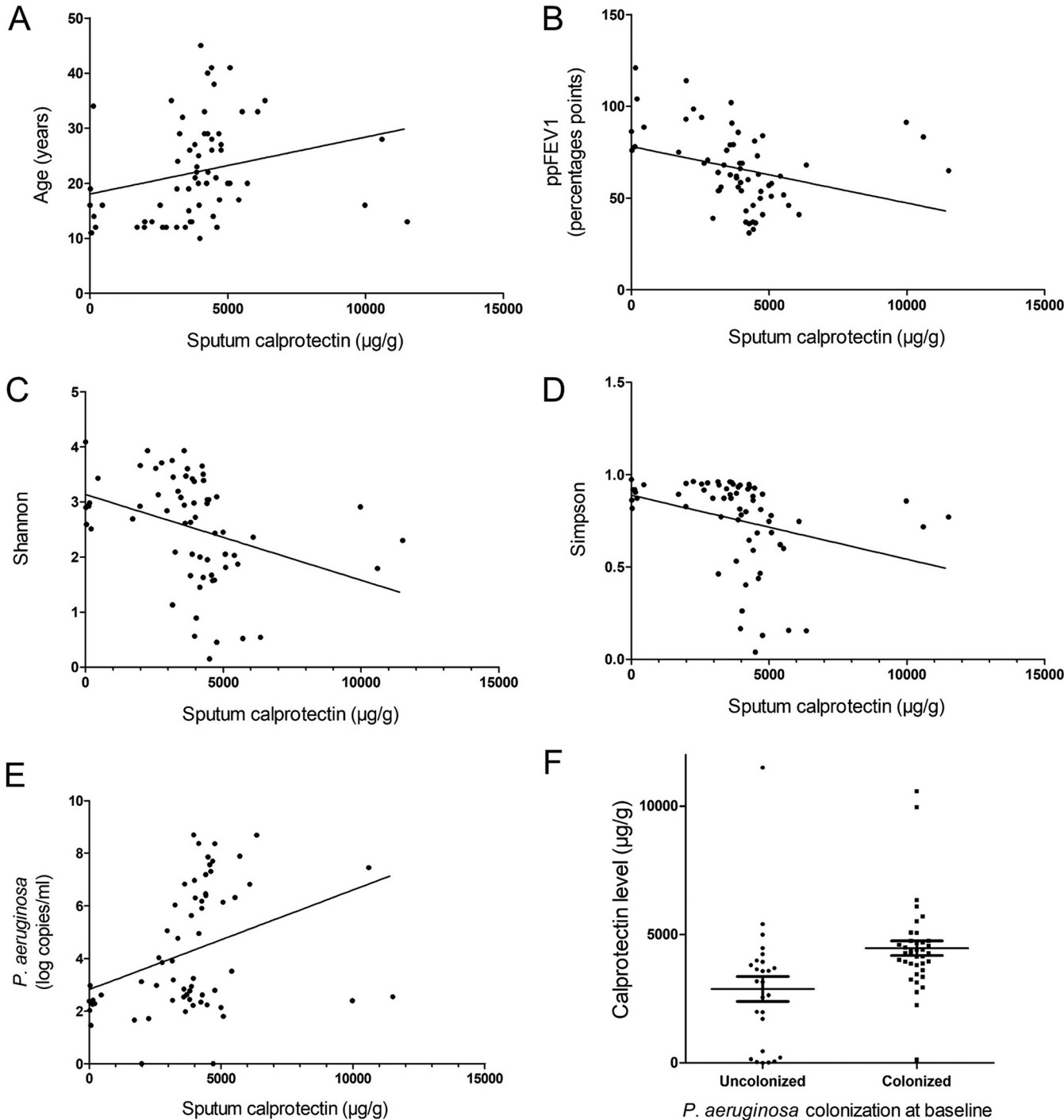

**FIG 2** Sputum calprotectin at baseline, according to patient and disease characteristics. Shown are sputum calprotectin levels at baseline according to age (A), ppFEV$_1$ (B), bacterial alpha-diversity indices (Shannon [C] and Simpson [D]), qPCR load of *P. aeruginosa* (E), and chronic colonization with *P. aeruginosa* (F).

different alpha- and beta-diversities between M0 and M6 in the population of 41 patients. No changes were observed in *P. aeruginosa* ASV relative abundances or *P. aeruginosa* quantitative PCR (qPCR) loads, bacterial and fungal loads, GM positivity rates, or sputum calprotectin levels (Table 2). Subgroup analyses by age (adolescents or adults) and lung function (ppFEV$_1$ of ≥80% or ppFEV$_1$ of <80%) at baseline also showed no significant changes in the airway microbiota and mycobiota with lumacaftor-ivacaftor treatment.

Chronic colonization with *P. aeruginosa* is a turning point in disease evolution. Patients

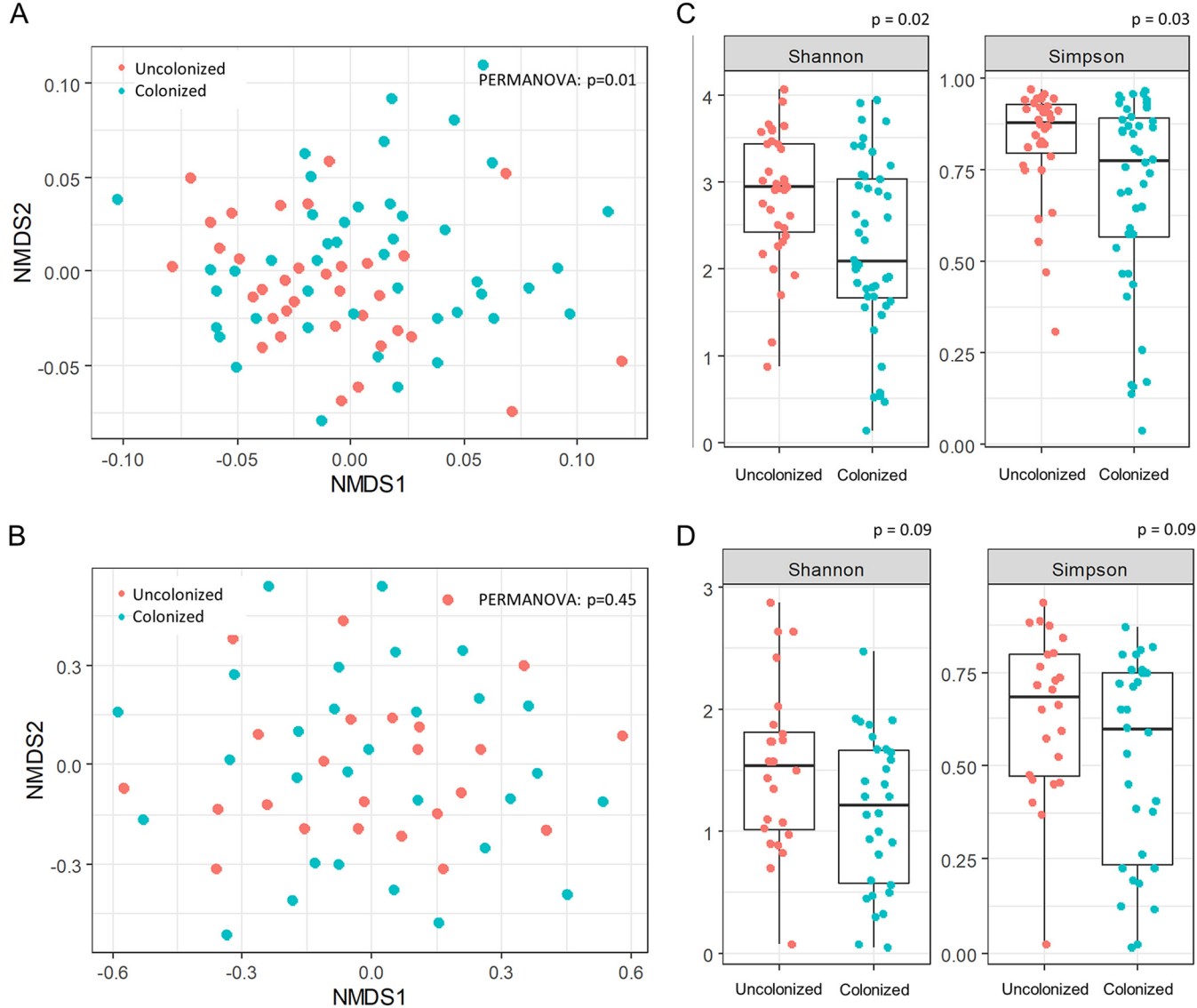

**FIG 3** Bacterial and fungal compositions of sputa at baseline according to the *P. aeruginosa* chronic colonization phenotype. (A and B) Comparison of targeted metagenomics data obtained from sputum samples at baseline (M0) between the patients colonized and those not colonized with *P. aeruginosa* for alpha- and beta-diversities. Beta-diversity, which assesses differences in microbial compositions between samples using a nonmetric multidimensional scaling (NMDS) ordination method with the Bray-Curtis distance metric of bacterial (A) and fungal (B) communities, is shown to measure the microbiota and mycobiota compositional similarity throughout. Permutational multivariate analysis of variance (PERMANOVA) was used to test sample clustering hypotheses. (C and D) Alpha-diversities of bacterial (C) and fungal (D) communities determined using the Shannon and Simpson indices.

chronically colonized with *P. aeruginosa* have a lower ppFEV$_1$ and more courses of antibiotics (Table S1). In addition, chronic colonization with *P. aeruginosa* was significantly associated with a higher calprotectin level and decreased bacterial alpha-diversity indices (Fig. 2F and Fig. 3C). Shifts in the bacterial community occurred more in patients not chronically colonized with *P. aeruginosa* than in patients colonized with *P. aeruginosa* (Fig. 4). Only patients without such colonization had significant increases in bacterial alpha-diversity indices under lumacaftor-ivacaftor treatment (Fig. 4). We did not observe any significant changes in fungal alpha-diversity, bacterial and fungal beta-diversities (data not shown), or calprotectin or microbial loads (Table 2). Regarding the M0-M6 microbiota and mycobiota data, DESeq2 analysis revealed a significant increase in *Malassezia restricta* and decreases in *Candida albicans*, *Capnocytophaga* spp., *Veillonella* spp., TM7x spp., *Rothia* spp., and *Fusobacterium* spp. (with no differences in evolution between Gram-positive and Gram-negative organisms) in patients not chronically colonized with *P. aeruginosa* (Table S2).

**TABLE 2** Evolution of clinical, microbial, and inflammatory parameters after 6 months of lumacaftor-ivacaftor treatment and according to *P. aeruginosa* chronic colonization status at baseline[a]

| | Value for group | | | | | | | | | | | |
|---|---|---|---|---|---|---|---|---|---|---|---|---|
| | All patients with M0 sputum samples (n = 75) | | | Patients with M0 and M6 sputum samples | | | | | | | | |
| | | | | All patients (n = 41) | | | Patients with *P. aeruginosa* chronic colonization (n = 27) | | | Patients not chronically colonized with *P. aeruginosa* (n = 14) | | |
| Parameter | Initiation | At 6 mo | P | Initiation | At 6 mo | P | Initiation | At 6 mo | P | Initiation | At 6 mo | P |
| Median ppFEV$_1$ (%) (IQR) | 65 (53, 84) | 65 (47, 91) | 0.1 | 58 (42, 78) | 62 (45, 85) | 0.3 | 51 (38, 63) | 47 (41, 66) | 0.3 | 79 (62, 93) | 86 (63, 94) | 0.3 |
| Median BMI (kg/m$^2$) (IQR) | 18.8 (17.3, 21.0) | 19.8 (18.2, 22.0) | **<0.001** | 19.2 (17.5, 20.3) | 19.6 (18.1, 21.7) | **<0.01** | 19.6 (17.8, 20.7) | 19.6 (18.3, 22.0) | 0.07 | 18.4 (17.1, 20.2) | 19.8 (17.8, 20.7) | **0.01** |
| Median BMI Z-score (IQR)[b] | −0.8 (−1.1, 0.1) | −0.3 (−0.8, 0.2) | 0.3 | −0.8 (−1.2, −0.1) | −0.31 (−0.8, 0.1) | 0.3 | −0.9 (−1.4, −0.5) | −0.52 (−0.9, −0.4) | 0.7 | −0.6 (−1.1, 0.3) | −0.2 (−0.3, 0.2) | 0.4 |
| No. (%) of patients with ≥1 i.v. antibiotic course[c] | 40 (56) | 29 (40) | **<0.01** | 25 (64) | 17 (42) | **0.01** | 21 (78) | 15 (56) | **0.04** | 4 (33) | 2 (15) | 0.5 |
| No. (%) of patients with pulmonary colonization | | | | | | | | | | | | |
| MSSA | 45 (60) | 31 (46) | 0.06 | 21 (51) | 14 (38) | 0.2 | 14 (52) | 7 (30) | 0.07 | 7 (50) | 7 (50) | 1 |
| MRSA | 12 (16) | 10 (15) | 1 | 8 (20) | 6 (16) | 1 | 5 (19) | 3 (13) | NA | 3 (21) | 3 (21) | 1 |
| H. influenzae | 7 (9) | 7 (10) | 1 | 3 (7) | 5 (14) | 0.7 | 1 (4) | 2 (9) | 1 | 2 (14) | 3 (21) | 1 |
| P. aeruginosa[d] | 41 (55) | 35 (50) | 0.7 | 27 (66) | 16 (48) | 1 | 27 (100) | 24 (89) | NA | 0 (0) | 2 (14) | NA |
| B. cepacia | 2 (3) | 1 (2) | NA | 1 (2) | 0 | NA | 1 | 0 | NA | 0 | 0 | NA |
| A. fumigatus[d] | 23 (39) | 23 (37) | 0.6 | 13 (46) | 22 (61) | 1 | 8 (44) | 11 (52) | 1 | 5 (50) | 5 (42) | 1 |
| Sputum dosage | | | | | | | | | | | | |
| Median calprotectin level (μg/mL) (IQR) | 3,941 (3,098, 4,581) | | | 4,232 (3,216, 4,652) | 4,018 (3,066, 4,675) | 0.5 | 4,413 (3,948, 4,760) | 4,361 (3,673, 4,812) | | 3,168 (2,141, 3,845) | 3,067 (799, 4,018) | 0.5 |
| No. (%) of patients with GM index of >1 | 19 (25) | | | 13 (32) | 17 (44) | 0.2 | 10 (37) | 13 (50) | 0.3 | 3 (23) | 4 (31) | 1 |
| Median total fungal load (log pg/μL) (IQR) | 0.7 (0, 1.3) | | | 0.7 (0, 1.3) | 0.7 (0, 1.4) | 0.6 | 0.7 (0, 1.2) | 0.8 (0, 1.3) | 0.4 | 0.7 (0.1, 1.3) | 0.5 (0.1, 1.3) | 1 |
| Median total bacterial load (log pg/μL) (IQR) | 2.3 (1.8, 2.8) | | | 2.1 (1.7, 2.6) | 2.4 (1.4, 2.7) | 0.8 | 2.1 (1.6, 2.6) | 2.4 (1.7, 2.7) | 0.4 | 2.2 (1.9, 2.8) | 2.3 (1.4, 2.6) | 0.8 |
| Median P. aeruginosa load (log copies/mL) (IQR) | 3.0 (2.4, 6.3) | | | 3.7 (2.4, 6.1) | 4.7 (2.6, 7.2) | 0.1 | 5.8 (3.4, 6.4) | 6.1 (3.7, 7.4) | 0.3 | 2.4 (2.4, 3.0) | 2.7 (2.4, 3.2) | 0.5 |

[a]Data are means (±SD), medians (IQRs), or numbers (percentages). ppFEV$_1$, percent predicted forced expiratory volume in 1 s; BMI, body mass index; MSSA, methicillin–susceptible *S. aureus*; MRSA, methicillin-resistant *S. aureus*; NA, not applicable. Significant values (p < 0.05) are in boldface.
[b]For adolescents.
[c]These criteria were evaluated over a period of 1 year prior to the M0 or M6 visit.
[d]*P. aeruginosa* and *A. fumigatus* colonizations refer to chronic colonization status defined according to local practices as the presence of *P. aeruginosa* isolates in 3 consecutive cultures with at least 1 month between the positive cultures during the previous 6 months (49) or as >50% of samples positive in the last 12 months (50) and as 2 sputum cultures positive for *A. fumigatus* during the last 12 months (51).

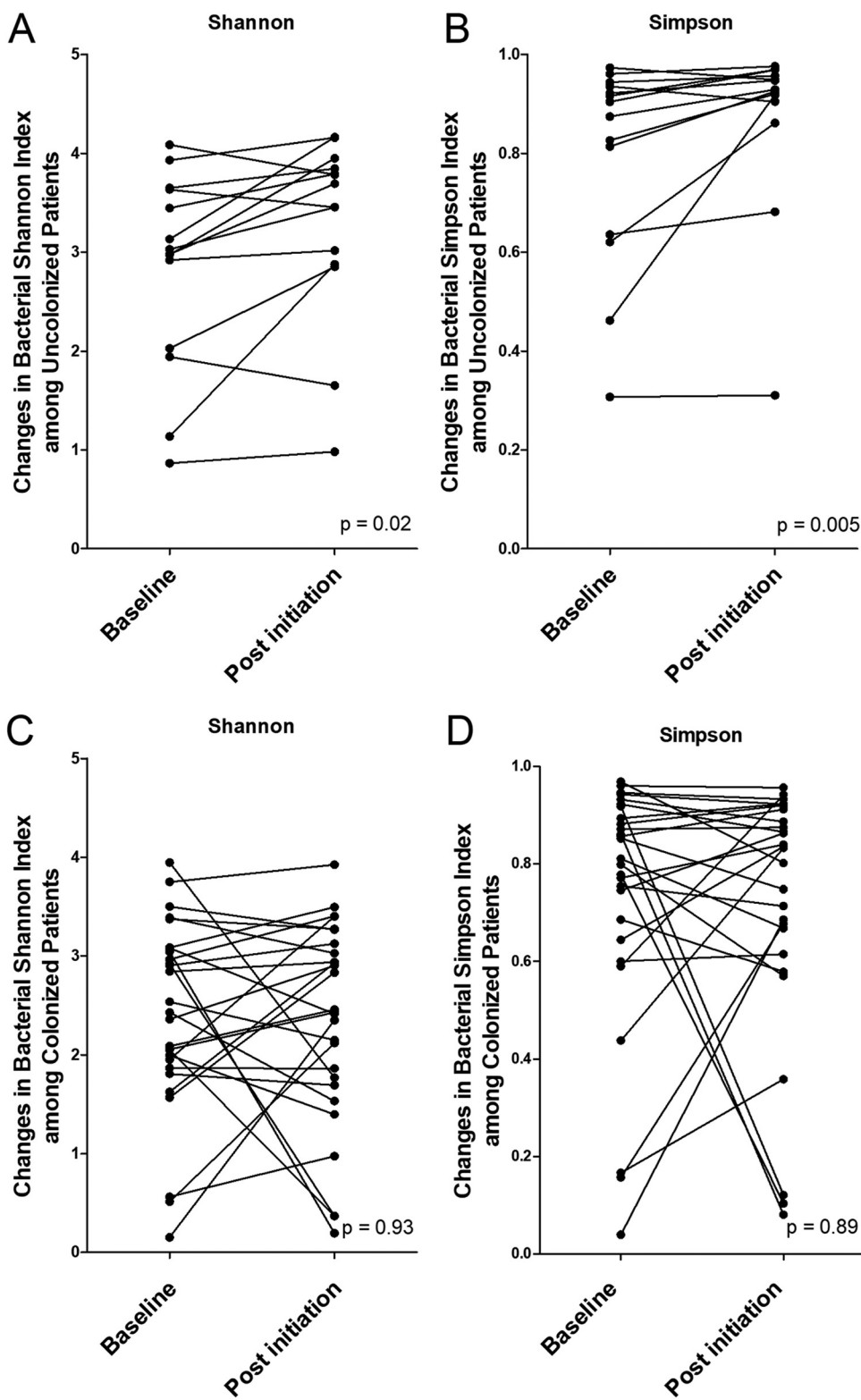

**FIG 4** Evolution of bacterial alpha-diversity indices with lumacaftor-ivacaftor treatment according to *P. aeruginosa* chronic colonization at baseline. Shown is the evolution of bacterial alpha-diversity indices with lumacaftor-ivacaftor treatment in patients without (A and B) and with (C and D) *P. aeruginosa* chronic colonization at baseline.

## DISCUSSION

We assessed the evolution of F508del homozygous patients treated with lumacaftor-ivacaftor over 6 months. First, the initiation of lumacaftor-ivacaftor was associated with improved BMI and decreased i.v. antibiotic courses. Second, changes in the airway microbiota under treatment were dependent on the status of chronic colonization by *P. aeruginosa* at baseline.

In our population of 75 CF patients, we did not observe a significant improvement in ppFEV$_1$ under treatment despite continuous exposure to lumacaftor-ivacaftor. In contrast, real-life data in a study on a large number of patients indicated a 3% increase in ppFEV$_1$ (10). However, ppFEV$_1$ evolution is multifactorial (i.e., local inflammation or the presence of bronchiectasis), as suggested by the partial correlation between sweat levels of chloride and ppFEV$_1$ evolution (12, 29, 30). In addition, improvements in ppFEV$_1$ with lumacaftor-ivacaftor are more difficult to establish in patients with good lung function (ppFEV$_1$ of ≥90%) or advanced lung disease (ppFEV$_1$ of <40%), with the latter condition representing one-third of our cohort values at baseline (11, 29, 30). Because the study was observational and sweat chloride or LCI estimates were not performed in routine care at 6 months of follow-up, we did not have data regarding these sweat parameters (which are more sensitive than ppFEV$_1$ [11, 31]). However, we observed improvements in BMI and intravenous antibiotic courses, which reflect more consistent effectiveness outcomes according to numerous studies (8, 10, 11, 29, 30). To date, studies on the effects of lumacaftor-ivacaftor treatment have focused mainly on conventional clinical and biological parameters, except for a recent study that analyzed the bacterial microbiota (11).

It is now well recognized that the lung ecosystem, encompassing resident microbial communities and the local inflammatory response, is strongly associated with the pathophysiology and outcomes of CF patients (5). Few data are available on the evolution of the airway microbiota under lumacaftor-ivacaftor or elexacaftor-tezacaftor-ivacaftor treatment, and there is no information on the airway mycobiota (11, 26, 32, 33). In this study, no significant changes in the airway microbiota and mycobiota were observed in the cohort of patients with spontaneously expectorated sputa at both time points, in contrast to preliminary results suggesting that lumacaftor-ivacaftor may increase bacterial alpha-diversity indices (32). We found no significant changes in the relative abundances or qPCR loads of fungal, bacterial, and *P. aeruginosa* biomasses. The data on the effects of lumacaftor-ivacaftor on bacteria or *P. aeruginosa* abundances are contradictory and inconclusive (26, 32).

At baseline, the airway microbiota and mycobiota compositions were consistent with those in the literature on CF (4, 33–35). These microbial communities are influenced by different factors related to CF evolution (i.e., pulmonary function and chronic colonization by *P. aeruginosa* or *A. fumigatus*), which may also affect changes in the microbial communities under modulator treatment. The status of chronic colonization by *P. aeruginosa* caught our attention because of its clinical relevance, its role in the evolution of the microbiota under modulator therapy (6, 24, 26, 32, 33), and its correlation with calprotectin levels in the sputum. In this study, patients who were not chronically colonized with *P. aeruginosa* at lumacaftor-ivacaftor treatment initiation had lower calprotectin levels before treatment and significant increases in bacterial alpha-diversity after 6 months of treatment. These results were congruent with the role of CF pathogens such as *P. aeruginosa*, which show decreased diversity with increased lung disease severity (28). With lumacaftor-ivacaftor, the same patients showed significant decreases in the abundances of anaerobes (i.e., *Fusobacterium* and *Veillonella*) as well as *Rothia*, which may contribute to lung damage through interactions with CF pathogens, including *P. aeruginosa* (6). These bacteria can degrade sputum mucins by generating short-chain amino acids and fatty acids that stimulate *P. aeruginosa* growth (36). The production of short-chain fatty acids by anaerobes could also be involved in the inflammatory response through the interleukin-8 (IL-8)-related pathway (6, 37). In addition, *P. aeruginosa* uses substrates produced by *Rothia* to generate primary metabolites

*in vitro* (5, 38). The role of anaerobes remains to be investigated by differentiating bacterial species and patient phenotypes based on the increased abundances of some anaerobes (*Porphyromonas* and *Peptostreptococcus*) in patients with a ppFEV$_1$ of ≥80% (33, 39).

Although the airway microbiota is relatively stable over time, even under ivacaftor treatment (31), the airway mycobiota has been shown to fluctuate over time, conditioned by the inhalation of environmental conidia (34). The impact of lumacaftor-ivacaftor treatment on the airway mycobiota is therefore difficult to analyze because its effect must be greater than spontaneous variation (40). However, in the patients who were not chronically colonized with *P. aeruginosa*, the *C. albicans* abundance was significantly decreased by lumacaftor-ivacaftor, and *C. albicans* is associated with worsening CF disease and a significant impairment of lung function (41). This result is in line with recent reports that *P. aeruginosa* coexists with *Candida* species more frequently in CF patients than in those with other respiratory disorders and that the rates of colonization by *Candida* and pathogenic bacteria in CF patients increase with age (42).

Because calprotectin is functionally related to neutrophil activation, we assessed the calprotectin level in the sputum supernatant to evaluate local inflammation (27, 43). We detected high levels of calprotectin, in line with the airway inflammation associated with lumacaftor-ivacaftor (11). In agreement with a previous report (24), we did not find a significant change in inflammation under treatment, even in patients who were not chronically colonized with *P. aeruginosa*, indicating that correcting airway inflammation in CF patients is challenging. Breath metabolome profiles were modified after 12 months of lumacaftor-ivacaftor therapy, in a way suggesting relationships with local inflammation and oxidative stress rather than with the airway microbiota composition (26).

This study had several limitations. Metabarcoding may not identify all microbial genera and species, although the V3-V4 and internal transcribed spacer 2 (ITS2) regions are efficient targets for amplifying bacterial and fungal DNAs, respectively (44, 45). To optimize the process, we used a denoising method (DADA2 with ASVs), which improves the resolution of low-frequency taxa and enhances the assessment of diversity compared to clustering methods (with operational taxonomic units) (46). Furthermore, patients without an M6 sputum sample were less chronically colonized with *P. aeruginosa* and had a higher ppFEV$_1$ at baseline. They can therefore be considered to have less severe and less advanced CF disease, which may render them less able to expectorate at 6 months, as recently suggested (11). Moreover, the success of CFTR modulators has resulted in several therapeutic options for chronic *P. aeruginosa* infection and subsequently requires adaptation to approaches for antimicrobial therapy (47). In addition, the evolution of the airway microbiota and mycobiota and inflammation may be different under lumacaftor-ivacaftor treatment, as proposed previously for ivacaftor treatment (24). Furthermore, the impact of lumacaftor-ivacaftor on the airway microbiota-mycobiota was evaluated after 6 months of treatment. Changes in the airway microbiota were observed from the first weeks of ivacaftor treatment (21), and a clinical response with lumacaftor-ivacaftor was achieved at 3 months (10), suggesting that evaluation at 6 months enables the detection of changes in the microbiota and mycobiota. However, this does not provide information on longer-term evolution, which may be different, as suggested by a previous study with ivacaftor (21). Therefore, longer-term studies are needed, taking into account confounding biases (poor compliance and antibiotic use) (22). Finally, because lumacaftor-ivacaftor treatment was part of standard care for F508del homozygous patients older than 12 years of age, it was not ethically conceivable to have control patients with CF paired for age, sex, and mutations. However, longitudinal analysis of the respiratory microbiota allowed the patients to be used as their own controls.

In conclusion, we report the evolution of the airway microbiota-mycobiota and inflammation in CF patients treated with lumacaftor-ivacaftor. In line with reports that alpha-diversity is a relevant marker of the microbial community (3, 48), our results highlight the importance of considering the CF lung ecosystem as a whole entity in

which the microbiome, the metabolome, and inflammation collectively contribute to disease progression (5, 26, 33). By combining clinical and biological parameters with inflammation and microbiota-mycobiota data, our findings support the notion that CFTR protein modulators should be started as early as possible, ideally before the patient is colonized with *P. aeruginosa* and the airways are irreversibly damaged (26). These data need to be confirmed, expressly by focusing on 2- to 11-year-old patients treated with lumacaftor-ivacaftor. Given our data and recent reports (11, 24, 26, 33), a comprehensive evaluation of CF management is warranted in the era of CFTR modulators, which should include clinical features, lung function evaluation (ppFEV$_1$, LCI, and MRI), and lung ecosystem analysis (local inflammation, microbiota-mycobiota, and metabolome assessments) in a personalized approach, together with assessments of the gut-lung axis, as recommended for ivacaftor (31).

## MATERIALS AND METHODS

**Study design and patients.** Patients enrolled in the LumIvaBiota longitudinal observational study were F508del homozygous and aged 12 years or older. They were monitored in one of six French CF centers that participated in the study (CRCM Centres of Bordeaux, Roscoff-Brest, Foch, Grenoble, Marseille, and Robert Debré). All patients initiated lumacaftor-ivacaftor treatment between December 2015 and June 2018 and expectorated spontaneously.

Sputum was obtained during care before treatment initiation (M0) and after 6 months (M6) of treatment. It consisted of spontaneous expectorations obtained during a visit to the hospital, eventually after physical therapy, according to the practice of each center. Participants were not required to perform an oral rinse or undergo hypertonic saline nebulization prior to expectoration. After the performance of conventional microbial analyses required for routine care, the remains of the spontaneously secreted sputa were frozen on-site at −20°C and shipped on dry ice to the Mycology Department of Bordeaux Hospital for centralized analyses of the microbiota-mycobiota, galactomannan (GM), and calprotectin. At both visits, patient clinical status was documented using data from the French National Prospective Cohort nested within the French Cystic Fibrosis Registry (10).

All patients, or their guardians as applicable, received information about the study. Nonopposition was obtained before the collection of the remains of sputum samples and clinical data (age, gender, ppFEV$_1$, BMI, medications, and results of microbial cultures of sputa performed during routine care). Microbiological data were collected to determine each patient's state of pulmonary colonization with *S. aureus*, *Haemophilus influenzae*, *Burkholderia cepacia*, *P. aeruginosa*, and *Aspergillus fumigatus*. Regarding *P. aeruginosa* and *A. fumigatus* colonization, chronic colonization was defined according to local practice as the presence of *P. aeruginosa* isolates in three consecutive cultures with at least 1 month between positive cultures during the previous 6 months (49) or as >50% of samples positive in the last 12 months (50) and as two sputum cultures positive for *A. fumigatus* during the last 12 months (51). Because there was no consensus definition for colonization with the other pathogens, the determination of colonization status was at the discretion of the investigator.

Samples were obtained from the Bordeaux Centre for Biological Collections (authorization number AC-2014-2166). This study was registered at ClinicalTrials.gov under identifier NCT03565692.

**Pretreatment of sputa and DNA extraction.** After pretreatment with an equal volume of Sputasol (Oxoid, Basingstoke, UK) for 30 min at 37°C followed by centrifugation (1,500 × *g* for 10 min), the supernatants and pellets were separately frozen at −20°C for calprotectin and GM assessments and DNA extraction, respectively (4, 52). We used the DNeasy PowerSoil kit (Qiagen, Les Ulis, France) to extract DNA from the samples, as described previously (53), after ensuring that it allowed the lysis of all bacteria and fungi in our artificial community (see the supplemental material). Next, we followed the manufacturer's protocol and enhanced the mechanical lysis step with Precellys evolution (two cycles of 30 s at 7,000 rpm), as described previously (54). Negative controls (250 $\mu$L of DNA-free water) were processed using the same protocol. DNA samples were used at 20 ng/$\mu$L.

**Microbiota and mycobiota evaluation using metabarcoding.** The taxonomic composition of sputa was assessed by targeting the V3-V4 and ITS2 regions of rRNA, as described previously (54). The primers used to amplify the V3-V4 and ITS2 loci were as follows: 16S-forward (TACGGRAGGCAGCAG), 16S-reverse (CTACCNGGGTATCTAAT), ITS2-forward (GTGARTCATCGAATCTTT), and ITS2-reverse (GATATG CTTAAGTTCAGCGGGT). In addition to the negative extraction controls, library blanks and two positive controls (in-house artificial bacterial and fungal communities [see the supplemental material]) were processed, sequenced alongside the patient samples, and used to validate the experimental procedures. PCR amplification was performed by using barcoded primers (final concentration of 0.2 $\mu$M) at an annealing temperature of 50°C for 30 cycles. PCR products were checked on an Agilent automated system, purified by using magnetic beads, and mixed in equimolar concentrations. Next-generation sequencing was performed using 250-bp paired-end technology on a MiSeq system (Illumina, San Diego, CA) at the Plateforme Génome Transcriptome de Bordeaux (PGTB) platform of Bordeaux University.

**Analysis of bacterial and fungal reads.** Reads were demultiplexed; primers were removed using CutAdapt. Samples were processed through the DADA2 pipeline in R (version 4.0.3) using standard filtering parameters, trimming, dereplication, and merging of paired-end reads (46, 55, 56). As recently proposed (57), only forward sequences were analyzed for characterizing the fungal community. Two distinct

amplicon sequence variant (ASV) tables were constructed, and taxonomy was assigned from the Silva database (release 138) for bacterial ASVs and from the Unite database (release 8.2) for fungal ASVs. We used mock communities to detect potential noneficient experiments and the negative controls to identify and remove potential reagent contaminants of the microbiota and mycobiota with the microDecon R package (58). ASVs present in fewer than 1% of the samples were removed (56). The coverage of each sample was assessed using rarefaction curves. All samples reached the plateau of rarefaction curves except for 10 samples from 8 patients with fewer than 100 fungal reads; these were removed from the final ITS2 analyses.

**Quantification of the bacterial, fungal, and *P. aeruginosa* loads.** qPCR targeting the 16S loci was used to quantify total bacterial loads, as previously described (11, 54). Quantification was performed using a standard range of *Escherichia coli* (ATCC 25922) concentrations (2.79 to 2,787.1 pg/$\mu$L).

*P. aeruginosa* abundance was quantified using a combination of two qPCRs (*oprL* and *ecfX-gyrB*), which have a sensitivity of 100% with a threshold of 10 CFU/mL and a specificity of 100% (59–61).

Fungal sequencing was based on amplification of the ITS2 region, consistent with published data showing better performance of ITS2 sequencing compared to other regions, including the 18S region (44). Published and widely used primers based on targeting of the fungal 18S region were selected for quantification of fungal load by qPCR (62). Quantification was performed using a standard range of *Candida albicans* (ATCC 5314) DNA concentrations (0.37 pg/$\mu$L to 3,663.5 pg/$\mu$L).

**Calprotectin dosages.** The levels of calprotectin, a proinflammatory factor abundant in CF sputum (27, 43), in the sputum supernatants were evaluated by an enzyme-linked immunosorbent assay (ELISA) (catalog number S100A8/9; Buhlmann Laboratories AG, Schonenbuch, Switzerland) (27, 43).

**Galactomannan dosages.** A Platelia *Aspergillus* enzyme immunoassay (EIA) (Bio-Rad, Marnes-La-Coquette, France) was used to assess the GM antigen indices of the sputum supernatants according to the manufacturer's recommended protocol; an index of $\geq$1 was considered positive, and assays with positive results were repeated in duplicate (63, 64).

**Statistical analyses.** Results are means ($\pm$standard deviations [SD]) for parametric variables, medians (interquartile ranges [IQRs]) for nonparametric variables or absolute values, and percentages for categorical variables. The nonparametric Wilcoxon-Mann-Whitney test was used to compare quantitative variables between groups. Correlations were calculated using the Spearman method. McNemar's test and the Wilcoxon signed-rank test were used to analyze paired nominal data and quantitative data, respectively. $P$ values were corrected for multiple testing using Benjamini-Hochberg adjustment (65).

For microbiota and mycobiota data, alpha-diversity metrics (Simpson and Shannon indices) were generated using the phyloseq R package. For cross-sectional analyses, at a specific time, significant differences in alpha-diversity indices were identified using the Wilcoxon rank sum test. For longitudinal analyses, the Wilcoxon signed-rank test was used. Differences in beta-diversities were tested by permutational multivariate analysis of variance (PERMANOVA) in the vegan R package with 10,000 permutations. DESeq2 (66) was used to perform two-class testing for differential relative abundances. Paired tests were used when comparing data before and after treatment initiation. The LEfSe method was used to identify metabarcoding biomarkers (67).

Analyses were performed in R studio (version 1.3.1056 for Windows); a $P$ value of $<$0.05 was considered indicative of statistical significance.

**Data availability.** The 16S rRNA gene and ITS2 sequences have been submitted to the European Nucleotide Archive (accession number PRJEB53549). The codes are available at https://github.com/raphaelenaud/LumIvaBiota. Other data sets generated and/or analyzed during the current study are not publicly available but are available from the corresponding author upon reasonable request.

## SUPPLEMENTAL MATERIAL

Supplemental material is available online only.
**SUPPLEMENTAL FILE 1**, PDF file, 0.9 MB.

## ACKNOWLEDGMENTS

We thank the patients and their families for their participation in the study as well as all nurses, physicians, and clinical research coordinators who were involved in the study.

The LumIvaBiota study group was composed of Sébastien Imbert, Fabien Beaufils, and Patrick Berger (Bordeaux University, INSERM U1045); Cécile Bébéar, Julie Macey, Frédéric Perry, and Pauline Gallet (Bordeaux University Hospital); Guillaume Simon (Bordeaux University Hospital, CIC 1401); Erwan Guichoux, Marie Massot, and Benjamin Tyssandier (PGTB, Pierroton); Muriel Cornet, Isabelle Pin, and Catherine Llerena (Grenoble University Hospital); Stephanie Gouriou (Brest University Hospital); Michele Gerardin and Patricia Mariani (Assistance Publique Hôpitaux de Paris, Hôpital Robert Debré); Sophie Ramel (Roscoff Fondation Ildys); Dominique Grenet and Emilie Cardot-Martin (Foch Hospital); Jean-Christophe Dubus, Nathalie Stremler-Le Bel, Mélisande Baravalle-Einaud, and Stephane Ranque (Marseille University Hospital); Emmanuel Mas, Marie Mittaine, and Sophie Cassaing (Toulouse University Hospital); and Nathalie Wizla, Caroline Thumerelle,

Dominique Turck, Séverine Loridant, and Anne-Sophie Deleplanque (Lille University Hospital).

R.E., F.L.-S., E.C., and L.D. designed the study. R.E., F.L.-S., E.C., S.B., M.F., P.-R.B., G.H.-A., and investigators from the LumIvaBiota study group contributed to data collection. R.E., F.L.-S., E.C., L.V.-S., J.G., T.S., M.N., G.H.-A., and L.D. contributed to data management and analysis. R.E., F.L.-S., E.C., L.V.-S., M.N., and L.D. performed the statistical analysis. R.E., G.H.-A., and L.D. wrote the first draft of the manuscript, which was revised and approved for important intellectual content by all authors. All authors approved the final version of the manuscript.

This work was funded by research grants from the Vertex Society and the Vaincre la Mucoviscidose Association. Neither the Vertex Society nor the Vaincre la Mucoviscidose Association played any role in the data collection and analysis or in the decision to submit the manuscript.

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
