## [Reviewer comments · Microbiology Spectrum]

Microbiology Spectrum

Effect of Lumacaftor-Ivacaftor on airway microbiota-mycobiota and inflammation in patients with cystic fibrosis appears to be linked to *Pseudomonas aeruginosa* chronic colonization

Raphaël Enaud, Florian Lussac-Sorton, Elena Charpentier, Lourdes Velo-Suárez, Jennifer Guiraud, Stéphanie Bui, Michael Fayon, Thierry Schaeverbeke, Macha Nikolski, LumIvaBiota Group, Pierre-Régis Burgel, Geneviève Héry-Arnaud, and Laurence Delhaes

Corresponding Author(s): Laurence Delhaes, CHU Bordeaux

Review Timeline:

Submission Date:	June 15, 2022
Editorial Decision:	September 12, 2022
Revision Received:	November 28, 2022
Editorial Decision:	December 19, 2022
Revision Received:	January 11, 2023
Editorial Decision:	January 27, 2023
Revision Received:	January 31, 2023
Accepted:	February 11, 2023

Editor: Silvia Cardona

Reviewer(s): Disclosure of reviewer identity is with reference to reviewer comments included in decision letter(s). The following individuals involved in review of your submission have agreed to reveal their identity: Katie Lynn Summers (Reviewer #3)

Transaction Report:

DOI: <https://doi.org/10.1128/spectrum.02251-22>

September 12, 2022

Dr. Raphaël Enaud
Centre Hospitalier Universitaire de Bordeaux
Bordeaux
France

Re: Spectrum02251-22 (Lumacaftor-Ivacaftor effect on CF lung mycobiota-microbiota and inflammation is driven by Pseudomonas aeruginosa colonization)

Dear Dr. Raphaël Enaud:

Thank you for submitting your manuscript to Microbiology. While we are committed to expediting revisions, this review took longer than expected. The reason was that one individual accepted to review the manuscript but did not submit their critique. I needed to secure another reviewer later. I apologize for the delay this issue has caused.

As you can see below, your article was reviewed by two experts in the field. Both found strengths but pointed out that important modifications are needed before the work is ready for publication. In particular, one reviewer feels that crucial negative controls are missing and has concerns regarding the DNA extraction technique being used. If you feel you can address these major points, we will be willing to consider a revised manuscript, which will be sent to a new round of reviews.

Link Not Available

Sincerely,

Silvia Cardona

Journals Department
Reviewer comments:

Reviewer #2 (Comments for the Author):

Enaud and colleagues investigated how Lumacaftor-Ivacaftor, a novel combination therapy that restores function to the CFTR protein in cystic fibrosis patients, affects the composition of microbial communities and inflammation in the diseased CF lungs. The authors sampled patients prior to the initiation of Lumacaftor-Ivacaftor therapy and 6-months post-therapy and used next-generation sequencing technology to interrogate bacterial and fungal communities and biochemical laboratory tests to measure inflammatory markers.

Please see the attached PDF file for my major and minor review comments.

Reviewer #3 (Comments for the Author):

In this paper the authors assess the effect of Lumacaftor-Ivacaftor on the lung microbiome in cystic fibrosis patients. The topic is of general interest and is well-written and clear to understand. This data has the potential to tell us about the overall microbiome alterations during disease progression under the treatment of targeted drugs such as LUM/IVA. Detailed notes below:

-L79: italicize bacterial name

-L88-89: grammar needs correcting

-L126: A little more information would be helpful. How were these frozen? Liquid nitrogen? Straight into freezer? -20C or -80C? Others must be able to replicate your work exactly.

-L148: Why was this kit chosen for DNA extraction? It is not recommended for fungal samples as it does not target the diverse cell structures of fungi. In fact, Qiagen created a DNeasy PowerSoil Pro kit specifically to assist with fungal isolations. DNA extraction techniques play a significant role in altering the composition of mycobiome studies.

-L170/174: If ITS2 sequences were targeted for high throughput sequencing, why were 18S primers used to quantify fungal load? Why the switch between primer targets for fungi?

-L215: These results are very interesting, but the overall effect is difficult to assess when negative controls are missing. For example, what about CF patients not receiving LUM/IVA? What about non-CF sputum samples? These controls would greatly increase the information that can be gleaned from this sequencing-based study.

-L342: Please expand on this topic. Aside from restating what was seen in the fungi, please address how this fits in the literature. Were these results expected? Why or why not?

-Figure 1: some labels are blurry and need resolution fixed and could be bigger so readers can see the labels. Figure 1B in particular needs to be larger or better resolution.

-Figure 2E and 2F: *P. aeruginosa* is not italicized

-Figure 3A, 3B, and S3B: The labels of the samples are not able to be read at all. Can these be made bigger?

-Figure S2 axis labels need better resolution

Staff Comments:

Preparing Revision Guidelines

Please return the manuscript within 60 days; if you cannot complete the modification within this time period, please contact me. If you do not wish to modify the manuscript and prefer to submit it to another journal, please notify me of your decision immediately so that the manuscript may be formally withdrawn from consideration by Microbiology Spectrum.

If your manuscript is accepted for publication, you will be contacted separately about payment when the proofs are issued;

please follow the instructions in that e-mail. Arrangements for payment must be made before your article is published. For a complete list of **Publication Fees**, including supplemental material costs, please visit our website.

Enaud and colleagues investigated how Lumacaftor-Ivacaftor, a novel combination therapy that restores function to the CFTR protein in cystic fibrosis patients, affects the composition of microbial communities and inflammation in the diseased CF lungs. The authors sampled patients prior to the initiation of Lumacaftor-Ivacaftor therapy and 6-months post-therapy and used next-generation sequencing technology to interrogate bacterial and fungal communities and biochemical laboratory tests to measure inflammatory markers. I have a number of comments that I feel will help to enhance the quality of the manuscript as it was difficult to follow in parts.

Major comments

As a general comment, I would recommend that the authors consider reviewing the grammar used in the manuscript to ensure that the reader can follow along easily and that all valid points are highlighted appropriately.

Lines 105-109: There really needs to be a stronger literature review and summary of similar studies that have looked at the effects of CFTR modulator therapies on lung microbiota. You list a number of these in the References section (e.g. Graeber et al. 2021, Boutin et al. 2019; Neerincx et al. 2021), but there should be some additional information in your introduction to really highlight the gap in knowledge in the field, the novelty of your study and its importance. What is currently known about the influence of CFTR correctors/modulators on lung microbiota? Are there significant shifts in diversity? Are they transient or prolonged? Is there an impact on the abundance of primary CF lung pathogens? What were issues with previous studies?

Line 122-123: I would appreciate some additional details on specimen collection here (or in an Appendix if necessary). Did any patients receive physical therapy prior to expectorating? Was the expectoration performed at home or in the clinic? Were they advised on how to collect the specimen? Was an oral rinse (e.g. saline mouthwash or gargle) used prior to specimen collection in order to evaluate the impact of oropharyngeal microbiota contamination of the expectorated sputa? How many days were there between sample collection and treatment initiation – please clarify if these were collected on the day of treatment?

Lines 214-222, 293: The authors highlight that there were both adolescent and adult patients included in the study, however no analyses account for this. I would also appreciate a deeper analysis based on disease stage – were there any disease related criteria used at the onset of study enrollment? Please comment in the methods (I recognize there are analyses based on FEV1 – please describe how the groups were defined). I would appreciate if there was some effort made to examine differences in adolescent and adult microbiota or those of individuals with early-moderate vs advanced disease given that we know that despite microbiota stability with disease progression there are fluctuations that may occur in adolescence/individuals with less severe pulmonary disease. Figure S2 is a good example of where this would be relevant.

Lines 260-263: The authors indicate that there was no significant difference in microbial diversity between pre- and mid-treatment specimens, including no changes in *P. aeruginosa* abundance, total bacterial or fungal loads among all patients. However, they claim in the same paragraph that the effects

of the CFTR modulator therapy are driven by *P. aeruginosa* colonization (which is also indicated in the manuscript title). I do not feel there is sufficient evidence to support this claim, particularly as there were 34 patient samples (45% of the initial dataset – which is a limitation not mentioned in the Discussion) that were not included in the mid-treatment M6 dataset as they were unable to produce sputum. I would recommend that the authors revisit this portion of the manuscript and clarify their statements (e.g. change the wording to reflect a non-*P. aeruginosa* dominated microbiome). There is also a mention that shifts in alpha diversity were noted among those patients that did not have *P. aeruginosa* isolated. Please comment (in the discussion) as to whether increased diversity is anticipated to be beneficial overall to patients (either hypothetically or with literature support) given what is known about the CF lung microbiome and disease state (e.g. perhaps changes in diversity are seen with *P. aeruginosa* more at the strain level vs an overall shift in abundance). Why might this not have been observed for patients with *P. aeruginosa*? Were the study patients still receiving their standard CF antimicrobial therapeutic regimens in addition to the modulator therapies? Please comment on this as it may also influence some of the data seen if so.

Line 282: Can the authors provide justification for the 6 month sampling time point and a lack of any additional sampling past this point?

Minor comments

Lines 88-95: Please consider revising this section as it does not clearly articulate the point you are trying to make. What factors drive the evolution of disease and shape the CF lung microbiota? How would these be different to what CFTR modulators may achieve?

Lines 96-97: Please comment on “traditional” therapeutic regimens that were administered pre-CFTR modulators as this will help to contextualize the importance of modulator therapy. How common is the F508del mutation (Please state this in the text)? I would also recommend commenting on the length of Lumacaftor-ivacaftor therapy (i.e. are they short-term or chronic therapies).

Line 118, 133: What clinical information was collected from your study patients? I would recommend commenting on this in the text.

Lines 136-143: What about other pathogens? In Tables 1 and 2 you separate these out into “pulmonary colonization” and “pulmonary chronic colonization”. Can you elaborate on how you distinguished these categories?

Lines 152-157; Lines 160-161: Please provide additional information regarding the sequencing preparation (e.g. paired end reads, amplicon size, other parameters) and analysis in an Appendix as it is not reproducible in its current state. What were your quality metrics – please state.

Lines 170-172: This may be better suited in the Results section along with a summary of the sequencing quality metrics

Lines 176-177: What concentration ranges were used here? Please comment on this.

Lines 181-187: Were the manufacturer's recommended protocols used? What controls were used and were any of the tests repeated? Were the positive Galactomannan results repeated in duplicate?

Lines 204-205: Were there any statistical corrections for multiple comparisons?

Lines 207-208: I would recommend having these codes available in GitHub or a similar online repository

Line 223: Can the authors revisit the manuscript and be consistent with the order of bacterial and fungal analyses throughout the text? I have highlighted this line as an instance where it switches from fungal-bacterial whereas in other parts of the text the analyses are mentioned bacteria-fungal.

Lines 231-241: Was there a difference in Gram positive vs Gram negative organisms with respect to abundance pre- or during therapy?

Line 246 – I would avoid using the term “uncolonized” and would prefer “not detected” or something similar, as it may be that it is at a level that is too low to detect by current methods

Line 248-249: Please tone down the wording in this comment – as it is not necessarily true, this is only a measure at baseline

Line 303, 368: Please use terms other than “morbi-mortality” and “a fortiori”

Line 570: Table 1 is confusing to me on first glance and is difficult to interpret. It is unclear to me which of the columns “Missing Values” refers to – is it all of the columns? Similarly, you list “all patients N=75” etc in the other columns despite there being missing data. Please clarify. Instead of Table 1 having a Missing values column, perhaps it would be better to have a “Number of patients with corresponding clinical data (N=)” column or something similar.

Line 629: Please add a breakdown of adolescent vs adult in Figure S1

Pr L Delhaes MD-PhD & Dr. Raphaël Enaud, MD-PhD
CRCM pédiatrique
University Hospital of Bordeaux
Place Amélie Raba Léon
33076 Bordeaux Cedex
France

Bordeaux, the 16th of November 2022

Dr. Silvia T Cardona
Editor of Microbiology Spectrum

Ref: Submission of **“Lumacaftor-Ivacaftor effect on CF lung mycobiota-microbiota and inflammation is driven by Pseudomonas aeruginosa colonization”** by Enaud *et al.*

Dear Editor,

Thank you for your interest on our work. Please find enclosed our revised manuscript (ID: Spectrum02251-22) **“Lumacaftor-Ivacaftor effect on CF lung mycobiota-microbiota and inflammation is driven by Pseudomonas aeruginosa colonization”** by Enaud *et al.*, that we are resubmitting.

Thank you also for the editorial and reviewer comments. We have revised the manuscript to address these comments. The corresponding modifications are visible using the "Track Changes" function in Microsoft Word and we added some comments. We would like to respond to the comments in detail as follows.

Reviewer 2 Comments:

Thank you for taking time to review the article. Please find our point-by-point answers.

Major comments

1. As a general comment, I would recommend that the authors consider reviewing the grammar used in the manuscript to ensure that the reader can follow along easily and that all valid points are highlighted appropriately.

We have revised the grammar of the manuscript in the hope that it will be easier to follow.

2. Lines 105-109: There really needs to be a stronger literature review and summary of similar studies that have looked at the effects of CFTR modulator therapies on lung microbiota. You list a number of these in the References section (e.g. Graeber et al. 2021, Boutin et al. 2019; Neerincx et al. 2021),

but there should be some additional information in your introduction to really highlight the gap in knowledge in the field, the novelty of your study and its importance. What is currently known about the influence of CFTR correctors/modulators on lung microbiota? Are there significant shifts in diversity? Are they transient or prolonged? Is there an impact on the abundance of primary CF lung pathogens? What were issues with previous studies?

We have added a paragraph to the introduction to better situate the context of this study (see lines 113-133)

3. Line 122-123: I would appreciate some additional details on specimen collection here (or in an Appendix if necessary). Did any patients receive physical therapy prior to expectorating? Was the expectoration performed at home or in the clinic? Were they advised on how to collect the specimen? Was an oral rinse (e.g. saline mouthwash or gargle) used prior to specimen collection in order to evaluate the impact of oropharyngeal microbiota contamination of the expectorated sputa? How many days were there between sample collection and treatment initiation – please clarify if these were collected on the day of treatment?

We have added these details in the material and method (see lines 148-152).

4. Lines 214-222, 293: The authors highlight that there were both adolescent and adult patients included in the study, however no analyses account for this. I would also appreciate a deeper analysis based on disease stage – were there any disease related criteria used at the onset of study enrollment? Please comment in the methods (I recognize there are analyses based on FEV1 – please describe how the groups were defined). I would appreciate if there was some effort made to examine differences in adolescent and adult microbiota or those of individuals with early-moderate vs advanced disease given that we know that despite microbiota stability with disease progression there are fluctuations that may occur in adolescence/individuals with less severe pulmonary disease. Figure S2 is a good example of where this would be relevant.

We have added the analyses at inclusion and for the evolution according to age in the results section, knowing that we have kept the choice to focus more on lung function than on age as a marker of disease severity (see lines 264-267; 281-283 and Figure S3)

5. Lines 260-263: The authors indicate that there was no significant difference in microbial diversity between pre- and mid-treatment specimens, including no changes in *P. aeruginosa* abundance, total bacterial or fungal loads among all patients. However, they claim in the same paragraph that the effects of the CFTR modulator therapy are driven by *P. aeruginosa* colonization (which is also indicated in the manuscript title). I do not feel there is sufficient evidence to support this claim, particularly as there were 34 patient samples (45% of the initial dataset – which is a limitation not mentioned in the Discussion) that were not included in the mid-treatment M6 dataset as they were unable to produce sputum. I would recommend that the authors revisit this portion of the manuscript and clarify their statements (e.g. change the wording to reflect a non-*P. aeruginosa* dominated microbiome). There is also a mention that shifts in alpha diversity were noted among those patients that did not have *P. aeruginosa* isolated. Please comment (in the discussion) as to whether increased diversity is anticipated to be beneficial overall to patients (either hypothetically or with literature support) given what is known about the CF lung microbiome and disease state (e.g. perhaps changes in diversity are seen with *P. aeruginosa* more at the strain level vs an overall shift in abundance). Why might this not have been observed for patients with *P. aeruginosa*? Were the study patients still receiving their standard CF antimicrobial therapeutic regimens in addition to the modulator therapies? Please comment on this as it may also influence some of the data seen if so.

As recommended, the authors revisit the title and clarify their statements, as follow:

- The title has been modified as “Lumacaftor-Ivacaftor effect on CF lung microbiota-mycobiota and inflammation appears to be linked to *Pseudomonas aeruginosa* chronic colonization”

- lines 295-297: section title has been changed as followed: “Changes in clinical outcomes, microbiota-mycobiota and inflammation characteristics under lumacaftor-ivacaftor treatment appears to be linked to *P. aeruginosa* chronic colonization status at baseline”
- lines 333-334: Discussion statements have been clarified.
- Regarding patients chronically colonized with *P. aeruginosa*, the authors have added comments in the discussion section (see lines 371-373; 414-416) as recommended.

6. Line 282: Can the authors provide justification for the 6 month sampling time point and a lack of any additional sampling past this point?

We discussed this point in the discussion section (see lines 418-426).

Minor comments

7. Lines 88-95: Please consider revising this section as it does not clearly articulate the point you are trying to make. What factors drive the evolution of disease and shape the CF lung microbiota? How would these be different to what CFTR modulators may achieve?

We have reviewed this section and further detailed the mechanisms of dysbiosis that may be involved in cystic fibrosis (see lines 88-91).

8. Lines 96-97: Please comment on “traditional” therapeutic regimens that were administered pre-CFTR modulators as this will help to contextualize the importance of modulator therapy. How common is the F508del mutation (Please state this in the text)? I would also recommend commenting on the length of Lumacaftor-Ivacaftor therapy (i.e. are they short-term or chronic therapies).

We have included all these details in the text (see lines 98-100 and line 102).

9. Line 118, 133: What clinical information was collected from your study patients? I would recommend commenting on this in the text.

We have included all these details in the text (see lines 160-161).

10. Lines 136-143: What about other pathogens? In Tables 1 and 2 you separate these out into “pulmonary colonization” and “pulmonary chronic colonization”. Can you elaborate on how you distinguished these categories?

These colonization criteria were recorded by the physician during the follow-up of each CF patients. According to local practices and published criteria, chronic colonization with *Pseudomonas aeruginosa* was defined as the presence of *P. aeruginosa* isolates in 3 consecutive cultures with at least one month between the positive cultures during the previous 6 months (Cantón R, et al. Antimicrobial therapy for pulmonary pathogenic colonization and infection by *Pseudomonas aeruginosa* in cystic fibrosis patients. *Clin Microbiol Infect Off Publ Eur Soc Clin Microbiol Infect Dis* 2005;11:690–703) or as more than 50% of samples positive in the last 12 months (Lee TWR, et al. Evaluation of a new definition for chronic *Pseudomonas aeruginosa* infection in cystic fibrosis patients. *J Cyst Fibros Off J Eur Cyst Fibros Soc* 2003;2:29–34). Chronic colonization with *Aspergillus fumigatus* was defined as 2 sputum cultures positive during the last 12 months (Amin R, et al. The effect of chronic infection with *Aspergillus fumigatus* on lung function and hospitalization in patients with cystic fibrosis. *Chest* 2010;137:171–176). Unfortunately, chronic definition for the other pathogens were not consensually defined.

The authors have clarified the material and method section (see new lines 163-169 and deletion of lines 148-155) and they simplify the tables 1 and 2 by grouping all the colonization.

11.Lines 152-157; Lines 160-161: Please provide additional information regarding the sequencing preparation (e.g. paired end reads, amplicon size, other parameters) and analysis in an Appendix as it is not reproducible in its current state. What were your quality metrics – please state.

The authors have provided additional information and stated on quality metric as follow: PCR products were checked on Agilent automated system (see lines 176, 182-183). As recommended in DADA2, we used standard filtering parameters [i.e. maxN=0 (DADA2 requires no ambiguous bases (Ns)), truncQ=2, rm.phix=TRUE and maxEE=2]. Of note, the maxEE parameter sets the maximum number of “expected errors” allowed in a read, which is a better filter than simply averaging quality scores. Of note: DADA2 incorporates quality information into its error model which makes the algorithm robust to lower quality sequence.

12.Lines 170-172: This may be better suited in the Results section along with a summary of the sequencing quality metrics

As recommended, the authors moved these data to the results section (see lines 190-191).

13.Lines 176-177: What concentration ranges were used here? Please comment on this.

We have specified the concentration ranges in the text (see lines 210-212).

14.Lines 181-187: Were the manufacturer’s recommended protocols used? What controls were used and were any of the tests repeated? Were the positive Galactomannan results repeated in duplicate?

The authors have detailed the use Galactomannan of kit (see lines 222-223).

15.Lines 204-205: Were there any statistical corrections for multiple comparisons.

Adjustments for multiple comparisons were made using a Benjamini–Hochberg correction. We have specified this point in the text (see lines 230-231).

16.Lines 207-208: I would recommend having these codes available in GitHub or a similar online repository

We have deposited the codes on Github and specified the address in the manuscript (<https://github.com/raphaelenaud/LumIvaBiota>) (see line 245).

17.Line 223: Can the authors revisit the manuscript and be consistent with the order of bacterial and fungal analyses throughout the text? I have highlighted this line as an instance where it switches from fungal- bacterial whereas in other parts of the text the analyses are mentioned bacteria-fungal.

We have homogenized this point in the manuscript.

18.Lines 231-241: Was there a difference in Gram positive vs Gram negative organisms with respect to abundance pre- or during therapy?

The authors compared relative abundances of bacterial ASVs with respect to pre- or during therapy; 10 bacterial species belonging to 4 Gram positive genus and 4 Gram negative genus expressed significant differences in relative abundances using DeSeq2 paired to species (see Table above).

baseMean	log2-Fold Change	lfcSE	stat	pvalue	padj	Phylum	Genus	Gram
516.9414	20.39942	3.877379	5.261137	0.0000001	0.0000087	Firmicutes	Streptococcus	Positif
875.9142	20.48885	3.855900	5.313636	0.0000001	0.0000087	Firmicutes	Veillonella	Négatif
191.4715	17.50008	3.633829	4.815879	0.0000015	0.0000596	Bacteroidota	Prevotella	Négatif
156.9243	14.79494	3.508353	4.217062	0.0000248	0.0007549	Firmicutes	Veillonella	Négatif
200.9114	-21.80327	5.354697	-4.071802	0.0000467	0.0010290	Proteobacteria	Haemophilus	Négatif
151.5757	14.67256	3.620350	4.052801	0.0000506	0.0010290	Actinobacteriota	Rothia	Positif
3482.8626	-19.52865	4.952285	-3.943362	0.0000803	0.0014003	Firmicutes	Staphylococcus	Positif
244.1629	-16.56228	4.267400	-3.881117	0.0001040	0.0015857	Firmicutes	Streptococcus	Positif
275.5114	-26.08639	7.147888	-3.649524	0.0002627	0.0035614	Bacteroidota	Porphyromonas	Négatif
109.4212	11.90627	4.078176	2.919509	0.0035058	0.0427711	Actinobacteriota	Atopobium	Positif

19.Line 246 – I would avoid using the term “uncolonized” and would prefer “not detected” or something similar, as it may be that it is at a level that is too low to detect by current methods

According to the new material & methods, the authors would prefer to modify this spelling and nuanced it by clearly referring to patients not chronically colonized with *P. aeruginosa* (instead of uncolonized patients).

20.Line 248-249: Please tone down the wording in this comment – as it is not necessarily true, this is only a measure at baseline

We have nuanced this point (see lines 290-291).

21.Line 303, 368: Please use terms other than “morbi-mortality” and “a fortiori”

We have changed these terms.

22.Line 570: Table 1 is confusing to me on first glance and is difficult to interpret. It is unclear to me which of the columns “Missing Values” refers to – is it all of the columns? Similarly, you list “all patients N=75” etc in the other columns despite there being missing data. Please clarify. Instead of Table 1 having a Missing values column, perhaps it would be better to have a “Number of patients with corresponding clinical data (N=)” column or something similar.

To avoid any confusion, we have retained in Table 1 only the clinical characteristics of all included patients and have added a table in supplementary data (Table S1) for the clinical characteristics of the subgroups (with and without sputum at M6).

23.Line 629: Please add a breakdown of adolescent vs adult in Figure S1

As these were not subgroups defined a priori in the study design for the analysis, we proposed not to include this information in the flow chart but specified it in the text and in Tables 1 and S1.

Reviewer #3

Thank you for taking time to review the article. Please find our point-by-point answers.

1. L79: italicize bacterial name

We have corrected this point.

2. L88-89: grammar needs correcting

We have corrected this point.

3. L126: A little more information would be helpful. How were these frozen? Liquid nitrogen? Straight into freezer? -20C or -80C? Others must be able to replicate your work exactly.

We have indicated the procedure for storing the samples (-20°C) (see line 153).

4. L148: Why was this kit chosen for DNA extraction? It is not recommended for fungal samples as it does not target the diverse cell structures of fungi. In fact, Qiagen created a DNeasy PowerSoil Pro kit specifically to assist with fungal isolations. DNA extraction techniques play a significant role in altering the composition of mycobiome studies.

One of the reasons for choosing this kit is that we are also studying the intestine-lung axis, and therefore we needed to have a kit common to both ecosystems, in order not to induce extraction biases. We first checked with fabricant that the PowerFecal Pro was also efficient for the lysis of fungi and we were also able to verify with the mocks that our process allowed to find the different expected fungal species.

5. L170/174: If ITS2 sequences were targeted for high throughput sequencing, why were 18S primers used to quantify fungal load? Why the switch between primer targets for fungi?

The authors selected ITS2 metabarcoding in agreement with published data that demonstrated a better performance of ITS2 amplification compared to other targets including 18S (Hoggard et al. Characterizing the Human Mycobiota: A Comparison of Small Subunit rRNA, ITS1, ITS2, and Large Subunit rRNA Genomic Targets. *Frontiers in Microbiology* 2018; doi: 10.3389/fmicb.2018.02208); this point has been detailed in lines 407-408.

On the other side, a published and widely used qPCR to perform fungal loads was based on targeting the fungal 18s region using a pair of primers and a TaqMan probe (C Liu et al. *FungiQuant: A broad-coverage fungal quantitative real-time PCR assay. BMC Microbiology* 2012, 12:255).

As this FungiQuant has a limit of quantification 25 copies and a limit of detection at 5 copies, we chosen to use it to quantify efficiently fungal loads in the present project.

6. L215: These results are very interesting, but the overall effect is difficult to assess when negative controls are missing. For example, what about CF patients not receiving LUM/IVA? What about non-CF sputum samples? These controls would greatly increase the information that can be gleaned from this sequencing-based study.

Since lumacaftor-ivacaftor treatment was part of standard care for deltaF508 homozygous patients older than 12 years, it was not ethically conceivable to have control patients with cystic fibrosis paired on age, sex, and mutations. However, longitudinal analysis of the respiratory microbiota allowed the patient to be taken for his own control. The authors have discussed this point in the discussion section.

7. L342: Please expand on this topic. Aside from restating what was seen in the fungi, please address how this fits in the literature. Were these results expected? Why or why not?

The authors have discussed this point lines 392-395.

8. Figure 1: some labels are blurry and need resolution fixed and could be bigger so readers can see the labels. Figure 1B in particular needs to be larger or better resolution.

We have made these changes

9. Figure 2E and 2F: *P. aeruginosa* is not italicized

We have made these changes

10. Figure 3A, 3B, and S3B: The labels of the samples are not able to be read at all. Can these be made bigger?

We improved the resolution of the axes and removed the names of the samples that were not essential and overloaded the figure.

11. Figure S2 axis labels need better resolution

We have improved the resolution of figure S2.

We hope that this revised version will now be found suitable for publication in *Microbiology Spectrum*.

Best regards,

Laurence Delhaes

Raphaël Enaud

December 19, 2022

Dr. Raphaël Enaud
Centre Hospitalier Universitaire de Bordeaux
Bordeaux
France

Re: Spectrum02251-22R1 (Lumacaftor-ivacaftor effect on CF lung microbiota-mycobiota and inflammation appears to be linked to Pseudomonas aeruginosa chronic colonization)

Dear Dr. Raphaël Enaud:

Thank you for submitting your manuscript to Microbiology Spectrum. As you will see your paper is very close to acceptance. As you will notice, one reviewer recommends improving clarity and removing the replication of data among Table 1,2 and S1. Please modify the manuscript along the lines the reviewers have recommended. As these revisions are quite minor, I expect that you should be able to turn in the revised paper in less than 30 days, if not sooner. If your manuscript was reviewed, you will find the reviewers' comments below.

When submitting the revised version of your paper, please provide (1) point-by-point responses to the issues raised by the reviewers as file type "Response to Reviewers," not in your cover letter, and (2) a PDF file that indicates the changes from the original submission (by highlighting or underlining the changes) as file type "Marked Up Manuscript - For Review Only". Please use this link to submit your revised manuscript. Detailed instructions on submitting your revised paper are below.

Link Not Available

Sincerely,

Silvia Cardona

Reviewer comments:

Reviewer #2 (Comments for the Author):

I thank the authors for considering my comments on the previous version of the manuscript. The manuscript has been improved, however, I still have some concerns. Please see the attached file for my additional comments.

Reviewer #3 (Comments for the Author):

Thank you to the authors for addressing my concerns. While not necessary, it might be nice to include a sentence or two in the methods describing the preliminary work done in choosing DNA extraction kits. You did appropriate preliminary work but the current manuscript does not provide the credit for it.

Preparing Revision Guidelines

Please return the manuscript within 60 days; if you cannot complete the modification within this time period, please contact me. If you do not wish to modify the manuscript and prefer to submit it to another journal, please notify me of your decision immediately so that the manuscript may be formally withdrawn from consideration by Microbiology Spectrum.

Review for Enaud et al. Lumafactor-Ivacaftor effect on CF lung microbiota-mycobiota and inflammation appears to be linked to Pseudomonas aeruginosa chronic colonization

I thank the authors for considering my comments on the previous version of the manuscript. The manuscript has been improved, however, I still have some concerns. Please see below.

Additional Comments

- 1) The manuscript still requires additional grammatical review (particularly in the Introduction and Discussion) as there are issues with flow and sentence structure throughout the manuscript which at times makes reading difficult.
- 2) Line 118 – I recommend changing the sentence that states “restoring a microbial composition closer to that of a healthy individual” to “restoring the pulmonary microbiota to that more closely resembling early disease”, given that the use of “healthy” is a bit subjective in this context.
- 3) Lines 280-282 – The authors state that some genera were “differentially expressed”. Please revise this statement to indicate that these were relative abundance measurements rather than measurements of gene expression. Similarly, clarify at the end of the sentence that these measurements were taken at baseline, as the sentence currently suggests that some analysis of change in abundance was performed here.
- 4) In my previous review I had commented that the authors should address whether there was a general overall change in the abundance of Gram positive or Gram negative organisms between M0- and M6-therapy samples (previous sentence was referred to on Lines 231-241; now Lines 328-334). The authors have provided a table in their response in which they state that 10 bacterial species (4 Gram negative genera and 4 Gram positive genera) had notable differences in relative abundances pre- and during-therapy. Can the authors please mention this in the text or add a clarifying sentence?
- 5) Line 298 – The statement that “this finding suggesting a clinical relevance of *P. aeruginosa* chronic colonization” provides an interpretation of results (e.g. Discussion) rather than a statement of results. Please ensure this is dealt with in the Discussion rather than Results.
- 6) Tables 1, Table 2, Table S1. In my previous review I commented that Table 1 was initially confusing and required some modification in order to clarify the data presented. The authors have revised this table and created an additional Table S1 which is included in the Supplemental info. The data provided in Table 1 is essentially repeated in both Tables S1 and Table 2 which is not necessary. This information should be stated in a single table to avoid repeating data (also be aware that different numbers, rounded to the nearest whole number vs decimals, are provided for some clinical measurements). This could all be collapsed into Table S1 and Table 2 kept in the main body. The authors also need to (i) include units for ppFEV1 and (ii) clarify what is meant by “missing values”, as there is no footnote given that explains this.

Pr L Delhaes MD-PhD & Dr. Raphaël Enaud, MD-PhD
CRCM pédiatrique
University Hospital of Bordeaux
Place Amélie Raba Léon
33076 Bordeaux Cedex
France

Bordeaux, the 04th of January 2023

Dr. Silvia T Cardona
Editor of Microbiology Spectrum

Ref: Submission of **“Lumacaftor-Ivacaftor effect on CF lung mycobiota-microbiota and inflammation is driven by Pseudomonas aeruginosa colonization”** by Enaud *et al.*

Dear Editor,

Thank you for your interest on our work. Please find enclosed our revised manuscript (ID: Spectrum02251-22) **“Lumacaftor-Ivacaftor effect on CF lung mycobiota-microbiota and inflammation is driven by Pseudomonas aeruginosa colonization”** by Enaud *et al.*, that we are resubmitting.

Thank you also for the editorial and reviewer comments. We have revised the manuscript to address these comments. The corresponding modifications are visible using the "Track Changes" function in Microsoft Word and we added some comments. We would like to respond to the comments in detail as follows.

Reviewer 2 Comments:

Thank you for taking time to review the article. Please find our point-by-point answers.

1) The manuscript still requires additional grammatical review (particularly in the Introduction and Discussion) as there are issues with flow and sentence structure throughout the manuscript which at times makes reading difficult.

We have submitted our manuscript to a certified English-speaking reviewer. Please find below the certificate of manuscript proofreading.

2) Line 118 – I recommend changing the sentence that states “restoring a microbial composition closer to that of a healthy individual” to “restoring the pulmonary microbiota to that more closely resembling early disease”, given that the use of “healthy” is a bit subjective in this context.

Thank you for this suggestion, which we have included in the manuscript.

3) Lines 280-282 – The authors state that some genera were “differentially expressed”. Please revise this statement to indicate that these were relative abundance measurements rather than measurements of gene expression. Similarly, clarify at the end of the sentence that these measurements were taken at baseline, as the sentence currently suggests that some analysis of change in abundance was performed here.

We have corrected this point, and rephrased the sentence with : “Among the 16 genera whose relative abundance was significantly different at baseline in patients with FEV1 \geq 80%, we observed an overrepresentation of *Streptococcus*, *Porphyromonas*, *Actinomyces*, *TM7x* and *Peptostreptococcus* abundances (Figure 1D).”

4) In my previous review I had commented that the authors should address whether there was a general overall change in the abundance of Gram positive or Gram negative organisms between M0- and M6-therapy samples (previous sentence was referred to on Lines 231-241; now Lines 328-334). The authors have provided a table in their response in which they state that 10 bacterial species (4 Gram negative genera and 4 Gram positive genera) had notable differences in relative abundances pre- and during-therapy. Can the authors please mention this in the text or add a clarifying sentence?

We have included this point by adding the concept of Gram to Table S3 (now Table S2) and modifying the results sentence with: “Comparing M0-M6 microbiota-mycobiota data, DESeq2 analysis revealed a significant increase of *Malassezia restricta* and a decrease of *C. albicans*, *Capnocytophaga* spp., *Veillonella* spp., *TM7x* spp., *Rothia* spp. and *Fusobacterium* spp. (without a difference evolution between Gram positive and negative organisms) in patients not chronically colonized with *P. aeruginosa* (Table S3 in the online supplement).

5) Line 298 – The statement that “this finding suggesting a clinical relevance of *P. aeruginosa* chronic colonization” provides an interpretation of results (e.g. Discussion) rather than a statement of results. Please ensure this is dealt with in the Discussion rather than Results.

We have removed this statement from the results section, as this point is already included in the discussion.

6) Tables 1, Table 2, Table S1. In my previous review I commented that Table 1 was initially confusing and required some modification in order to clarify the data presented. The authors have revised this table and created an additional Table S1 which is included in the Supplemental info. The data provided in Table 1 is essentially repeated in both Tables S1 and Table 2 which is not necessary. This information should be stated in a single table to avoid repeating data (also be aware that different numbers, rounded to the nearest whole number vs decimals, are provided for some clinical measurements). This could all be collapsed into Table S1 and Table 2 kept in the main body. The authors also need to (i) include units for ppFEV1 and (ii) clarify what is meant by “missing values”, as there is no footnote given that explains this.

We combined Table 1 and S1, specified the missing values for each column, and added a footnote to define "missing values". We added the unit for ppFEV1 and checked each number.

Reviewer #3

Thank you for taking time to review the article. Please find our answers.

Thank you to the authors for addressing my concerns. While not necessary, it might be nice to include a sentence or two in the methods describing the preliminary work done in choosing DNA extraction kits. You did appropriate preliminary work but the current manuscript does not provide the credit for it.

We have rephrased the sentence in the methods with : “We have chosen the DNeasy PowerSoil kit (Qiagen, Les Ulis, France) to extract DNA from the samples, after ensuring that it allows the lysis of all bacteria and fungi in our artificial community (see supplementary data). We then followed the manufacturer's protocol, by enhancing the mechanical lysis step with the use of the Precellys evolution (2 cycles of 30 s at 7000 rpm), as previously described”.

We hope that this revised version will now be found suitable for publication in *Microbiology Spectrum*.

Best regards,

Laurence Delhaes

Raphaël Enaud

Textcheck Certificate

Refnum:	23010510
Title:	Lumacaftor-Ivacaftor effect on CF lung microbiota-mycobiota and inflammation appears to be linked to Pseudomonas aeruginosa chronic colonization
Date:	2023/01/10

We hereby certify that Textcheck has checked and corrected the English in the manuscript named above.

A specialist editor with suitable professional knowledge (M.Sc. or Ph.D./M.D.) reviewed and corrected the English. An English language specialist subsequently checked the paper again. The first language of both editors is English.

Please direct any questions regarding this certificate or the English in the certified paper to: certified@textcheck.com
(Please quote our reference number: '23010510')

January 27, 2023

Dr. Raphaël Enaud
Centre Hospitalier Universitaire de Bordeaux
Bordeaux
France

Re: Spectrum02251-22R2 (Effect of Lumacaftor-Ivacaftor on airway microbiota-mycobiota and inflammation in patients with cystic fibrosis appears to be linked to *Pseudomonas aeruginosa* chronic colonization)

Dear Dr. Raphaël Enaud:

Thank you for submitting your manuscript to Microbiology Spectrum. As you will see your paper is very close to acceptance. Please modify the manuscript along the lines I have recommended. As these revisions are quite minor, I expect that you should be able to turn in the revised paper in less than 30 days, if not sooner. You will find the reviewers' comments to address below.

When submitting the revised version of your paper, please provide (1) point-by-point responses to the issues raised by the reviewers as file type "Response to Reviewers," not in your cover letter, and (2) a PDF file that indicates the changes from the original submission (by highlighting or underlining the changes) as file type "Marked Up Manuscript - For Review Only". Please use this link to submit your revised manuscript. Detailed instructions on submitting your revised paper are below.

Link Not Available

Sincerely,

Silvia Cardona

Pending comments:

- 1) Please, explain the rationale for the switch between ITS and 18S primers. Please add a sentence to the methods to explain your rationale (you provided it in response to reviewers but not in the paper).
- 2) Please, explain why the DNA extraction kit was chosen in the methodology section.
- 3) There are portions of the manuscript that differ between the "marked up" and "clean" versions, particularly in the Discussion section. Please, ensure that the marked-up document and the clean version are the same.

Preparing Revision Guidelines

- Point-by-point responses to the issues raised by the reviewers in a file named "Response to Reviewers," NOT IN YOUR COVER LETTER.

- Upload a compare copy of the manuscript (without figures) as a "Marked-Up Manuscript" file.
- Each figure must be uploaded as a separate file, and any multipanel figures must be assembled into one file.
- Manuscript: A .DOC version of the revised manuscript
- Figures: Editable, high-resolution, individual figure files are required at revision, TIFF or EPS files are preferred

Please return the manuscript within 60 days; if you cannot complete the modification within this time period, please contact me. If you do not wish to modify the manuscript and prefer to submit it to another journal, please notify me of your decision immediately so that the manuscript may be formally withdrawn from consideration by Microbiology Spectrum.

Pr L Delhaes MD-PhD & Dr. Raphaël Enaud, MD-PhD
CRCM pédiatrique
University Hospital of Bordeaux
Place Amélie Raba Léon
33076 Bordeaux Cedex
France

Bordeaux, the 30th of January 2023

Dr. Silvia T Cardona
Editor of Microbiology Spectrum

Ref: Submission of “**Lumacaftor-Ivacaftor effect on CF lung mycobiota-microbiota and inflammation is driven by *Pseudomonas aeruginosa* colonization**” by Enaud *et al.*

Dear Editor,

Thank you for your interest on our work. Please find enclosed our revised manuscript (ID: Spectrum02251-22) “**Lumacaftor-Ivacaftor effect on CF lung mycobiota-microbiota and inflammation is driven by *Pseudomonas aeruginosa* colonization**” by Enaud *et al.*, that we are resubmitting.

We have revised the manuscript to address editorial comments. The corresponding modifications are visible using the "Track Changes" function in Microsoft Word and we added some comments. We would like to respond to the comments in detail as follows.

Editorial Comments:

1) Please, explain the rationale for the switch between ITS and 18S primers. Please add a sentence to the methods to explain your rationale (you provided it in response to reviewers but not in the paper).

We have revised the relevant part of the methods section to explain our rationale:

“qPCR targeting the 16S loci was used to quantify total bacterial loads as previously described (11, 33). Quantification was performed using a standard range of *Escherichia coli* (ATCC 25922, 2.79 to 2787.1 pg/μL).

P. aeruginosa abundance was quantified using a combination of two qPCRs (of oprL and ecfX/gyrB), which have a sensitivity of 100% with a threshold of 10 CFU/mL and a specificity of 100% (39–41).

While the fungal metabarcoding was based on ITS2 amplification, in agreement with published data showing better performance compared to other targets including 18S (42), a published and widely used qPCR based on targeting the fungal 18s region was chosen to perform fungal loads (43). Quantification was performed using a standard range of *Candida albicans* DNAs (ATCC 5314, 0.37 pg/μL to 3663.5 pg/μL).”

2) Please, explain why the DNA extraction kit was chosen in the methodology section.

We had already included this justification in the last version submitted, but these changes were not visible in the "marked up" version. We have added a reference to justify this choice.

“We used the DNeasy PowerSoil kit (Qiagen, Les Ulis, France) to extract DNA from the samples, as described previously (Ghuneim, LA.J., Raghuvanshi, R., Neugebauer, K.A. et al. Complex and unexpected outcomes of antibiotic therapy against a polymicrobial infection. *ISME J* 16, 2065–2075 (2022). <https://doi.org/10.1038/s41396-022-01252-5>), after ensuring that it allowed lysis of all bacteria and fungi in our artificial community (see Supplementary Data).

3) There are portions of the manuscript that differ between the "marked up" and "clean" versions, particularly in the Discussion section. Please, ensure that the marked-up document and the clean version are the same.

We apologize for the confusion. At the last submission, the “clean” version of the manuscript was the right one, while the “marked up” version was not updated. We therefore propose to submit a version highlighting the changes of the last two revisions.

We hope that this revised version will now be found suitable for publication in *Microbiology Spectrum*.

Best regards,

Laurence Delhaes

Raphaël Enaud

February 1, 2023

Dr. Raphaël Enaud
Centre Hospitalier Universitaire de Bordeaux
Bordeaux
France

Re: Spectrum02251-22R3 (Effect of Lumacaftor-Ivacaftor on airway microbiota-mycobiota and inflammation in patients with cystic fibrosis appears to be linked to Pseudomonas aeruginosa chronic colonization)

Dear Dr. Raphaël Enaud:

Your manuscript has been accepted, and I am forwarding it to the ASM Journals Department for publication. You will be notified when your proofs are ready to be viewed.

Sincerely,

Silvia Cardona
Editor, Microbiology Spectrum
